# Deciphering response dynamics and treatment resistance from circulating tumor DNA after CAR T-cells in multiple myeloma

Hitomi Hosoya[1,8], Mia Carleton[2,8], Kailee Tanaka[2], Brian Sworder[3], Shriya Syal[4], Bita Sahaf[4], Alisha M. Maltos[5], Oscar Silva[5], Henning Stehr[5], Vanna Hovanky[1], George Duran[2], Tian Zhang[6], Michaela Liedtke[6], Sally Arai[1], David Iberri[6], David Miklos[1], Michael S. Khodadoust[2], Surbhi Sidana[1]✉ & David M. Kurtz[2,7]✉

Despite advances in treatments, multiple myeloma (MM) remains an incurable cancer where relapse is common. We developed a circulating tumor DNA (ctDNA) approach in order to characterize tumor genomics, monitor treatment response, and detect early relapse in MM. By sequencing 412 specimens from 64 patients with newly diagnosed or relapsed/refractory disease, we demonstrate the correlation between ctDNA and key clinical biomarkers, as well as patient outcomes. We further extend our approach to simultaneously track CAR-specific cell-free DNA (CAR-cfDNA) in patients undergoing anti-BCMA CAR T-cell (BCMA-CAR) therapy. We demonstrate that ctDNA levels following BCMA-CAR inversely correlate with relative time to progression (TTP), and that measurable residual disease (MRD) quantified by peripheral blood ctDNA (ctDNA-MRD) was concordant with clinical bone marrow MRD. Finally, we show that ctDNA-MRD can anticipate clinical relapse and identify the emergence of genomically-defined therapy-resistant clones. These findings suggest multiple clinical uses of ctDNA for MM in molecular characterization and disease surveillance.

Therapeutic advances including novel immunotherapies have improved outcomes of patients with multiple myeloma (MM)[1–4]. Despite this, most patients eventually experience disease relapse, and subsequent therapies become increasingly less effective. Disease biology also undergoes dramatic change through sequential gain of genetic mutations, copy number alterations, and epigenetic modifications. These changes lead to clinical manifestations of extramedullary disease and/or oligo/non-secretory disease, which pose challenges in clinical disease monitoring[5–8]. Chimeric antigen receptor (CAR) T-cell therapy demonstrates great promise in patients who

otherwise have dismal outcomes from conventional therapy. However, virtually all patients relapse eventually, and there are few established biomarkers predictive of clinical outcomes[9,10], and the mechanisms of resistance are largely unknown. Thus, there is an unmet need to identify a predictive biomarker, and to understand the mechanisms of resistance to this novel immunotherapy.

Currently, assessment of genomic aberrations including MRD is performed using bone marrow (BM) aspirate, thus limiting its frequency and utility. Furthermore, MM is a heterogeneous disease with the involvement of different clones in different anatomical sites[11],

[1]Division of Blood and Marrow Transplant and Cell Therapy, Department of Medicine, Stanford University, Stanford, CA, USA. [2]Division of Oncology, Department of Medicine, Stanford University, Stanford, CA, USA. [3]Division of Hematology/Oncology, Department of Medicine, University of California, Irvine CA, USA. [4]Center for Cell Therapy, Stanford Cancer Institute, Stanford University, Stanford, CA, USA. [5]Department of Pathology, Stanford University, Stanford, CA, USA. [6]Division of Hematology, Department of Medicine, Stanford University, Stanford, CA, USA. [7]Stanford Cancer Institute, Stanford University, Stanford, CA, USA. [8]These authors contributed equally: Hitomi Hosoya, Mia Carleton. ✉e-mail: surbhi.sidana@stanford.edu; dkurtz@stanford.edu

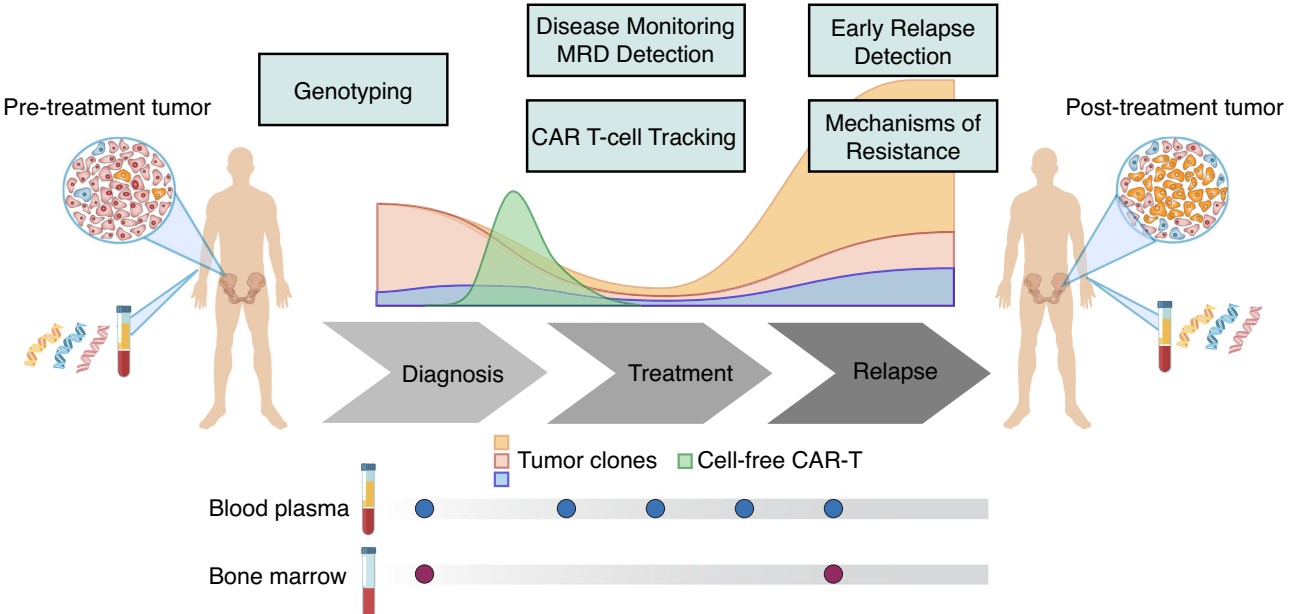

**Fig. 1 | Potential ctDNA application in multiple myeloma.** Schematic depicting the usage of cell-free DNA assay for disease characterization, treatment response monitoring, CAR T-cell tracking, and early relapse detection. By following sequential samples over the course of treatments, clonal dynamics, as well as treatment resistance mechanisms can be revealed.

which poses challenges for assessing genomic information of the whole tumor. Therefore, improved tools to overcome spatial heterogeneity and monitor disease are crucial for developing personalized therapeutic approaches.

Circulating tumor DNA (ctDNA) has emerged as a promising minimally invasive assay for diagnosing and monitoring various cancers, including several hematologic malignancies[12–15]. As ctDNA is shed into the blood from tumor cells throughout the body carrying tumor-related genomic information, it can be distinguished from non-tumor-derived cell-free DNA (cfDNA) and provides multifaceted genomic information from the whole tumor. Moreover, ctDNA has myriad potential uses, including molecular genotyping, treatment response assessment, and quantification of MRD using a blood draw. Various methodologies have been used to assess ctDNA including initial studies in MM[16–21]. However, these studies focus on either whole exome sequencing (WES) with relatively limited depth of sequencing, or targeted deep sequencing with a small number of genes, limiting the number of genetic alterations captured per case and the utility for applications in MRD[22,23]. Improved approaches are therefore greatly needed.

To address these unmet needs, we developed a robust and ultra-sensitive CAPP-Seq (cancer personalized profiling by deep sequencing)[24,25] hybrid capture approach for ctDNA in MM. Our approach allows genomic characterization, disease burden quantification, treatment response monitoring, and assessment of mechanisms of therapeutic resistance (Fig. 1). We demonstrate the potential applications of ctDNA in patients treated with CAR T-cell therapies to enable MRD detection, outcome prediction, and understanding of genomic determinants of therapeutic resistance.

## Results

### Developing CAPP-Seq for MM ctDNA detection
To profile genomic alterations in MM from ctDNA, we designed a targeted CAPP-Seq panel for MM utilizing publicly available WGS and WES data[26–29], including well-characterized coding genes harboring frequent mutations in MM. Furthermore, as the sensitivity for detecting residual tumor-derived DNA depends on having multiple mutations available per case[25,30], we included frequently observed hypermutated immunoglobulin loci and other non-coding genomic regions identified in previously reported WGS data[29], which had higher rates of mutations compared to canonical drivers mutations in coding regions (Fig. 2A). These regions had a significant overlap with diffuse large B-cell lymphoma (DLBCL) cases (Fig. S1) where these mutations were demonstrated as targets of activation-induced cytidine deaminase (AID)[31]. As AID is also indicated in aberrant somatic hypermutation in MM[32], the inclusion of these regions allows us to maximize the sensitivity for ctDNA-MRD detection from plasma[25,30]. This panel is described in Table S1.

We applied CAPP-Seq to bone marrow and blood plasma samples from 64 patients with plasma cell disorders including newly diagnosed multiple myeloma (NDMM, $n = 11$), treatment-naïve monoclonal gammopathy of undetermined significance (MGUS, $n = 3$), treatment-naïve smoldering myeloma (SMM, $n = 3$), and relapsed/refractory MM (RRMM, $n = 47$) (Table 1). A median of 83 single-nucleotide variants (SNVs, range 0–246) at an average allele frequency (AF) of 2.7% (range 0–51%) were detected per case. Compared to solid (lung and esophageal) malignancies where CAPP-Seq has been previously applied and established as a useful tool for tracking ctDNA-MRD, we detected a greater number of SNVs for tumor tracking in MM[33–35] (Fig. 2B, C), which directly implies an improved sensitivity compared to these diseases. Indeed, while previous studies in DLBCL identified a higher number of SNVs per case compared to MM, we did observe a similar level of ctDNA in pretreatment samples (Fig. 2C), indicating that ctDNA is a viable analyte in MM.

Importantly, our approach identifies and tracks more mutations than approaches focused on coding genes alone, thus increasing the sensitivity for ctDNA-MRD detection. To demonstrate this, we compared the performance of our panel with an in-house hematological mutational assay covering 164 genes (Stanford Tumor Actionable Mutation Panel for Hematopoietic and Lymphoid Neoplasms; Heme-STAMP) performed on the BM for cases where paired data was available ($n = 11$). We saw significantly higher numbers of detected SNVs by our panel compared to Heme-STAMP, suggesting that a current clinical next-generation sequencing-based assay focusing on coding mutations does not detect an adequate number of mutations (Fig. 2D). Correspondingly, the limit of detection (LOD) significantly improved

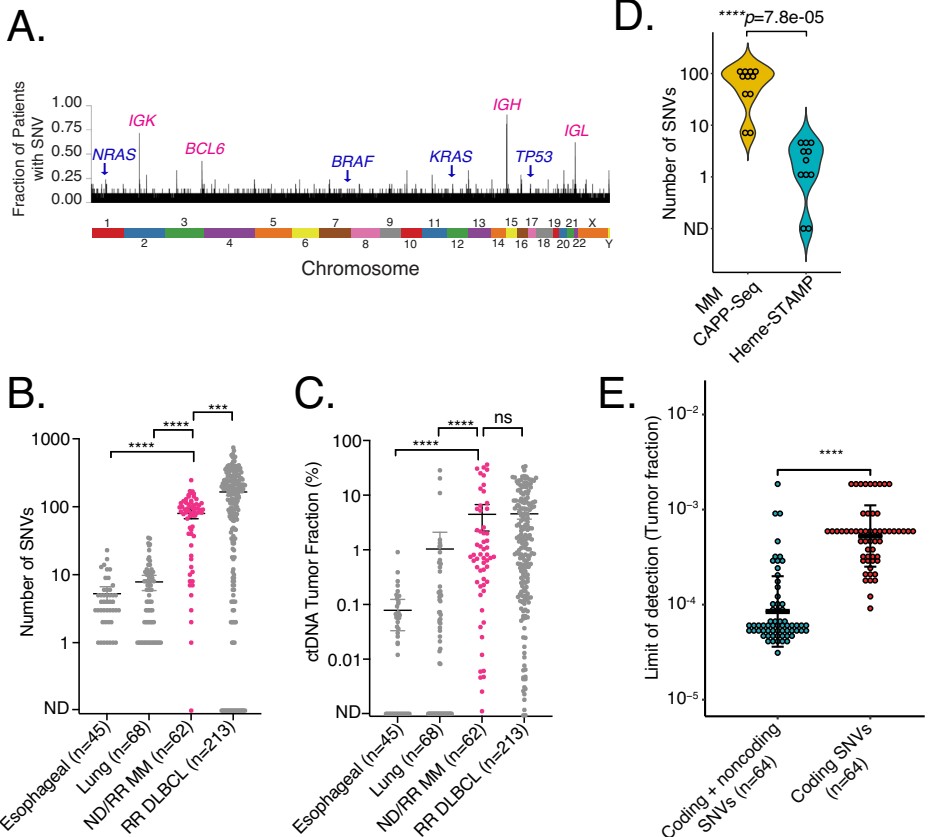

**Fig. 2 | Single nucleotide variants in multiple myeloma. A** A bar plot showing the distribution of single nucleotide variants in multiple myeloma from WGS data[29]. The genome was divided into 1000-bp bins, and the fraction of patients with a variant in each bin was calculated. Both frequently mutated driver coding genes (navy) and most frequently mutated genomic regions (pink) are highlighted. **B**, **C** Number of SNVs identified from genotyping samples (tumor for esophageal and lung cancers; tumor or blood plasma for MM or DLBCL), as well as allele frequencies of ctDNA from pre-treatment samples in order to demonstrate ctDNA detection in MM. Data are presented as mean values ± 95% CI. The *p*-values were calculated using a two-tailed Wilcoxon rank sum test. *p*-values were **B** 8.2e−14 (esophageal vs MM), 2e-15 (lung vs MM), 0.0003 (DLBCL vs MM); **C** 1.4e-9 (esophageal vs MM), 1.8e-8 (lung vs MM), 0.28 (DLBCL vs MM). ****P < 0.0001, ***P < 0.001, and P > 0.0001. **D** Number of SNVs identified by our CAPP-Seq assay compared to clinical NGS assay for hematological malignancies for samples that had paired data in our cohort (*n* = 11). The *p*-value was calculated by a two-sided Wilcoxon rank-sum test. **E** The ctDNA detection limits are shown for coding genes only and coding + non-coding genomic regions based on the number of mutations detected in our cohort. Data are presented as mean values ± SD. The *p*-value was calculated using the Wilcoxon rank sum test, showing 6.9e-16.

with increasing number of SNVs detected by incorporating non-coding genomic regions, as opposed to considering coding SNVs alone (Fig. 2E). Importantly, we saw a high degree of concordance (92%, 11/12 mutations) in the overlapping regions of the genome between these two panels, indicating the robustness of our genotyping approach, as well as implying that the increased number of mutations seen is driven by selecting enriched regions of the genome (Table S2). We additionally evaluated the specificity of our assay for both mutational genotyping and monitoring ctDNA-MRD, thus establishing the performance of panel-wide genotyping and the 95% specificity of our monitoring assay (Fig. S2, Supplemental Materials).

**Genomic characterization of MM via ctDNA**

We genotyped the disease of each patient in our cohort via bone marrow aspirate or cfDNA derived from blood plasma (Fig. 3A). 63 cases (98%) of cases had detectable SNVs. Among detected SNVs, 73% were in immunoglobulin loci (*IGH*, *IGK*, and *IGL*), 20% were in noncoding genomic regions, and 7% were in non-Ig coding genes (Fig. 3B), consistent with our expected enrichment of AID-related mutations. Indeed, detected mutations in noncoding genomic regions include intronic areas of *BCL6* (73% of patients), *LPP/BCL6* super-enhancer region (48%), and *BCL7A* (36%), which is concordant with previous reports[29,36,37]. Mutations in immunoglobulin loci were detected in 90% (*IGH*), 84%

(*IGL*), and 64% (*IGK*) of patients, respectively. Single base substitution mutational signatures in these immunoglobulin and noncoding regions demonstrated canonical signatures related to AID activity (SBS84 and SBS85), as well as non-canonical signatures previously associated with AID activity (SBS37, SBS39)[31,38] (Fig. 3C). The prevalence of AID-related mutations demonstrates that our approach captures biologically relevant mutations in addition to conventionally reported driver genes, increasing the number of mutations available to track MRD and thereby increasing sensitivity for disease detection. By using conventionally reported driver genes, only 45/64 cases (70%) had detected mutations. The improved performance for disease monitoring considering both coding and non-coding alterations is highlighted in a case vignette (Fig. 3D). This case had developed oligo-secretory disease with high levels of ctDNA, with quantitative levels tracking with treatment response. Notably, after ctDNA levels declined due to disease response, even when coding genes became undetectable (red arrows in Fig. 3D) at a point of low disease burden, multiple mutations in non-coding and AID-related regions remained detectable. In addition to improvements in disease detection, tracking non-coding variants allows assessment of clonal heterogeneity between ctDNA and bone marrow, where detected clonal composition evolves over time. This effect is highlighted by tracking SNVs in non-coding regions in blood plasma (Fig. S3).

**Table 1 | Demographics of patients included in the study**

|  | Median (range) or *N* (%) |
|---|---|
| Age (year) | 65 (30–81) |
| Sex, female | 24 (38%) |
| **Diagnosis** | |
| Newly diagnosed | 17 (27%) |
| Relapsed/refractory | 47 (73%) |
| **Type of disease** | |
| Multiple myeloma | 54 (84%) |
| MGUS/SMM | 6 (9%) |
| Hx of plasma cell leukemia | 4 (6%) |
| **Type of therapy** | |
| CAR-T cell therapy | 36 (56%) |
| Other therapy/observation | 28 (44%) |
| **R-ISS** | |
| I | 13 (20%) |
| II/III | 41 (64%) |
| unknown | 10 (16%) |
| High-risk cytogenetics* | 30 (47%) |
| Prior lines of therapy (median) | 5 (0–16) |
| History of extramedullary disease | 33 (52%) |
| Oligo/non-secretory disease | 8 (13%) |

*MGUS* monoclonal gammopathy of undetermined significance, *SMM* smoldering myeloma, *R-ISS* revised international staging system.
*High-risk cytogenetics include del(17p), t(4;14), t(14;16), gain(1q).

In addition to these immunoglobulin genes and non-coding mutations, we identified frequent SNVs in coding genes including *KRAS* (23% of patients), *NRAS* (14%), BRAF (13%), and *TP53* (16%), which is concordant with prior reports[27,28,39]. Of note, the rate of *TP53* mutation was significantly higher than in Walker et al.[39]. However, this potentially reflects the fact that Walker et al. only included NDMM, as opposed to our cohort, which also contains RRMM (Figs. 3E and S4 and S5).

**Validation of ctDNA single nucleotide variants and copy number alterations**
To assess the performance of blood plasma as a surrogate for tumor mutational genotyping, we compared mutations observed in the blood plasma with paired BM samples in 22 patients where BM samples were available and evaluable. The correlation between the allele fraction of each SNV detected in the BM and cfDNA is shown in Fig. S6, where 17/22 (77%) of cases demonstrated a significant positive correlation between tumor and plasma. For these samples, among SNVs detected in the BM, 60% (1360/2266) were also detectable in the plasma (Fig. 3F). As expected, we also observed that the sensitivity for detecting BM-derived SNVs improved as the burden of tumor DNA in the plasma increased (Fig. S7A). Overall, in samples where the mean ctDNA fraction was ≥0.3%, the sensitivity of detecting BM-derived SNVs was 82%. The frequency of overlapping mutations suggests that in most cases of MM, where the ctDNA amount is above this level, BM-derived SNVs are reliably detected from the plasma. In addition to BM-derived SNVs, we detected an additional 483 SNVs in the plasma across these 22 patients that were not present in the BM, including *KRAS*, *NRAS*, *TP53*, and *BRAF* mutations, suggesting ctDNA may capture spatial tumor heterogeneity not present in BM biopsies. While our assay does include deep sequencing of matched normal to exclude germline alleles and clonal hematopoiesis, it is possible these alleles could arise from another malignancy. However, none of these patients had second malignancies documented at the time of assessment, making this an unlikely source of these mutations. When considering mutations that were shared between BM and plasma vs those private to plasma, mutations in driver genes such as *NRAS*, *KRAS*, and *TP53* were more likely to be shared compared with other mutations,

suggesting these mutations may occur in early pathogenesis and are more clonal (Fig. S7B).

Given the importance of structural variants in MM disease biology, we also assessed the performance of our assay for identifying copy number alterations (CNAs) directly from the blood plasma. Detection of CNAs from plasma has been historically challenging, with prior methods focused on low-pass WGS reporting successful genotyping of arm-level alterations only at tumor fractions greater than 3%[40]. Here, we employed our previously described method, CANARy (Copy number ANomaly Assessment and RecoverY), a computational framework to use on- and off-target sequencing reads from targeted sequencing data to identify genome-wide copy number alterations (Fig. S8A)[34,41]. We compared the performance of CANARy for identifying CNAs with historical clinical fluorescence in situ hybridization (FISH) from BM for key CNAs (del(17p), gain(1q) and del(13q)). As with SNV genotyping, we observed that the sensitivity of detection improved as plasma AF increased. For samples with an AF ≥ 2.5%, CANARy demonstrated sensitivity of 80%, 80%, and 100% for detecting del(17p), gain(1q), and del(13q), respectively (Fig. S8B, C). This performance is comparable to dedicated LP-WGS methods from cfDNA[42], suggesting that CNAs detected using CANARy are concordant with BM FISH tests and that sensitivity improves with higher plasma tumor burden. Additionally, unlike FISH assay, which requires specific probes for chromosome regions of interest, CANARy assesses cfDNA CNAs across the whole genome, leading to the discovery of additional alterations. This analysis can be potentially useful in the research and discovery settings; however, we note that this assay should not replace the current gold standard bone marrow-based FISH test for clinical assessment.

**ctDNA levels are associated with adverse risks and disease burden of myeloma**
We next quantified ctDNA levels (Supplemental Methods) in patients with plasma cell neoplasms and correlated with clinical disease features. Among 64 patients, there were 70 samples that were collected at diagnosis or at relapse. Among them, ctDNA was detected in 96% (67/70) with a median of 2.0 log10 haploid genome equivalents per mL of plasma (hGE/mL). Levels of ctDNA were higher in MM compared to MM-precursors such as MGUS or SMM (2.10 vs 1.09 log hGE/mL, p = 0.028; Fig. 4A), as well as RRMM compared to NDMM (2.17 vs 1.15 log he/mL, *p* = 0.043; Fig. 4B) and patients with a history of extramedullary disease compared to those without the extramedullary disease (2.24 vs 1.41 log he/mL, *p* = 0.026; Fig. 4C). There was no significant difference if a patient had R-ISS I vs II/III (*p* = 0.050), oligo/non-secretory disease (*p* = 0.96), or high-risk cytogenetics (*p* = 0.058) in our cohort (Fig. S9A–C). We next assessed ctDNA levels correlating with clinical disease burden measurements. We observed a significant correlation between ctDNA and M-spike (rho = 0.55, *p* = 1.4 × 10^{-6}; Fig. 4D), as well as free light chain difference (dFLC) (rho = 0.69, *p* = 2.2 × 10^{-16}; Fig. 4E). Notably, in patients with oligo/non-secretory disease, where the blood-based assessment of disease burden is challenging clinically, we detected comparable ctDNA levels to patients with the secretory disease (1.3 vs 1.6 logs hGE/mL, *p* = 0.18 Figs. 4D and S9B).

**ctDNA dynamics to track treatment response in myeloma**
Having demonstrated the prognostic performance of baseline ctDNA levels, we next explored the utility of ctDNA to predict therapeutic responses. We considered 31 patients with a subsequent sample 28 days after initiation of therapy where clinical evaluation according to International Myeloma Working Group (IMWG) criteria was also available (CAR T, *n* = 21; non-cellular therapy, *n* = 10). We evaluated whether the change in ctDNA level on day 28 (D28) from pre-treatment correlated with the best response (Fig. 4F). The treatments for patients in this cohort are summarized in Table S3. The log-fold change (LFC) in ctDNA at D28 was significantly greater in patients who achieved a partial response (PR) or better, compared to patients with less than a

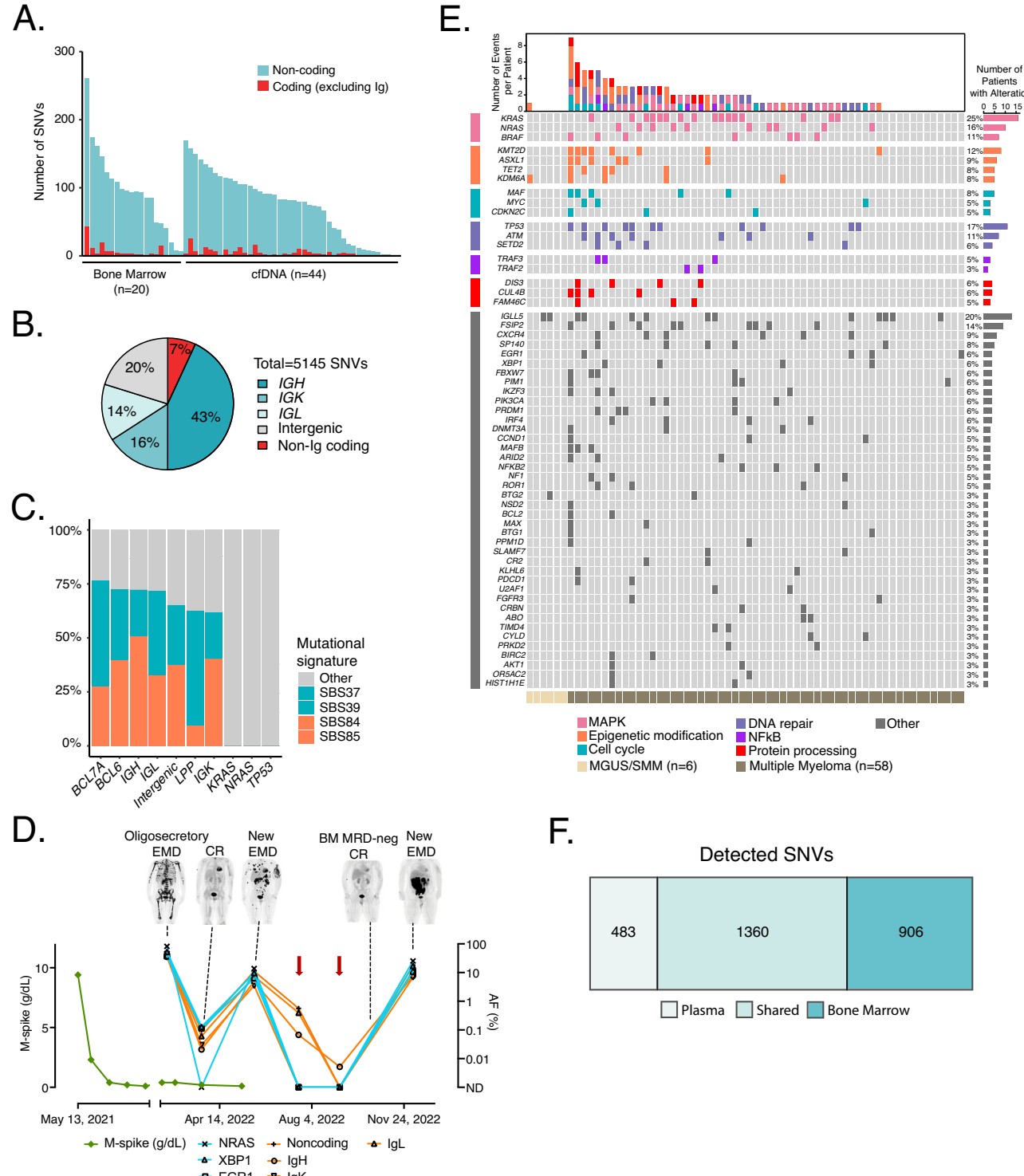

**Fig. 3 | Profiling of pre-treatment samples by CAPP-Seq. A** A bar plot depicting the number of SNVs detected from genotyping samples (bone marrow aspirate, $n = 20$; or plasma, "cfDNA", $n = 44$). The number of non-coding single-nucleotide variants (SNVs), inclusive of immunoglobulin (Ig) loci, are shown in aqua; the number of coding SNVs not including Ig loci are shown in red. **B** A pie-chart showing the breakdown of a total of 5145 SNVs detected in Fig. 3A. IGH immunoglobulin heavy chain, IGK immunoglobulin kappa light chain, IGL immunoglobulin lambda light chain. **C** A bar plot showing the mutational signatures of frequently detected intronic genes (BCL7A, BCL6, LPP/BCL6 super-enhancer region), immunoglobulin loci, and intergenic regions as compared to frequently detected coding mutations in driver genes (KRAS, NRAS, TP53). SBS37, 39, 84, and 85 are highlighted as being associated with AID activity. **D** An example of tracking coding and non-coding mutations for disease monitoring in a case. Red arrows point to timepoints when coding genes became undetectable but non-coding regions and immunoglobulin loci remained detectable. **E** An oncoprint showing the coding mutations detected in our cohort ($n = 64$). Only coding SNVs detected in at least two patients are shown. MGUS/SMM vs MM is identified by the color bar at the bottom. MAPK, mitogen-activated protein kinase; NF-κB, nuclear factor-κB. Source data are provided as a Source Data file. **F** Venn diagram showing concordance of single nucleotide variants detected from bone marrow (BM) and plasma across 22 paired samples.

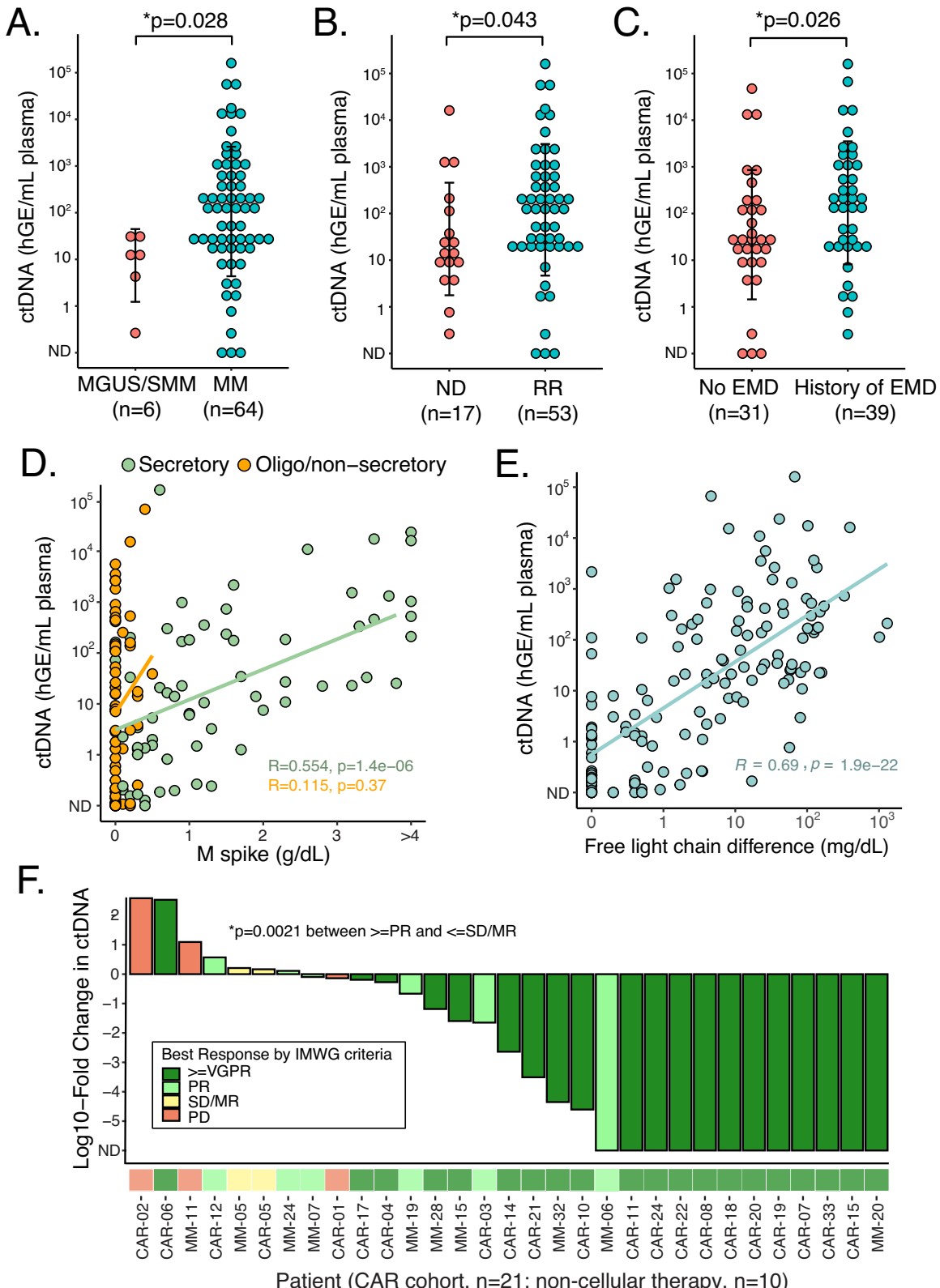

PR (median LFC −2.1 vs 0.03, $p = 0.002$). Interestingly, one significant outlier to this trend was observed (CAR-06), achieving a very good partial response (VGPR) by clinical criteria on D28, while observing a more than 100-fold increase in ctDNA. This patient relapsed on D82 after CAR with the new extramedullary disease, demonstrating that in some cases, ctDNA may serve as a more accurate assessment of treatment response compared to standard criteria. M-spike and difference in free light chains (dFLC) were also correlated with disease

response in this cohort, although numerous cases were unevaluable by one or both of these methods (Fig. S10).

## ctDNA levels are prognostic for time to progression after CAR T-cell therapy in myeloma

We then applied CAPP-Seq to serial samples from 36 patients with RRMM undergoing anti-BCMA CAR T-cell (BCMA-CAR) therapy. Patients were treated as part of the standard of care and received

**Fig. 4 | Correlation of ctDNA with clinical characteristics and therapeutic response. A–C** Levels of ctDNA grouped by disease type. MGUS monoclonal gammopathy of undetermined significance, SMM smoldering myeloma, MM multiple myeloma, ND newly diagnosed, RR relapsed/refractory, EMD extramedullary disease, hGE/mL human genome equivalent per mL of plasma, ND in x-axes, non-detected. Data are presented as mean ± SD. p-values were calculated by a two-sided Wilcoxon rank-sum test. **D, E** Correlation of ctDNA levels and clinical biomarkers (M-spike and difference between involved and uninvolved free light chains) by linear regression based on secretory disease status. Correlation coefficients and p-values were calculated using a two-tailed Spearman's rank-order test. **F** Waterfall plot of quantified molecular response (log10-transformed fold-change in ctDNA from day 0 to day 28 post-treatment). Cases are categorized and color-coded according to the best response as assessed by IMWG criteria. The p-value was calculated by a two-sided Wilcoxon rank-sum test. VGPR very good partial response, PR partial response, SD stable disease, MR minimal response, PD progressive disease.

---

either ide-cel ($n = 30$) or cilia-cel ($n = 6$) (Table S4). Genotyping of each patient's myeloma was performed using BM or blood plasma. We then tracked ctDNA-MRD levels and mutational profiles in serial samples for each patient throughout the course of therapy (Supplemental Methods). 31 of 36 (86%) patients had sufficient SNVs for evaluation and ctDNA monitoring. Notably, among the five patients with insufficient mutations for evaluation, four were assessed for clonal immunoglobulin sequences from BM by next-generation sequencing (NGS; ClonoSeq, Supplemental Methods). Three out of four patients failed to identify any dominant clone, demonstrating low disease burden. This highlights the importance of identifying mutations from cfDNA or BM at diagnosis or relapse when there is measurable disease for further disease monitoring and therapeutic response.

The median time to progression (TTP) and follow-up of the evaluable study cohort ($n = 31$) were 6.6 and 22.2 months, respectively; 23 patients experienced disease progression during follow-up. We categorized patients into those with progression within 90 days of CAR infusion (early progressors) and those without progression in that timeframe (non-early progressors). 12/18 (67%) of non-early progressors cleared ctDNA by a median of 21 days post-CAR, whereas only 1/8 (13%) early progressors had undetectable ctDNA by D28 (Fig. 5A). On D28, non-early progressors had significantly lower ctDNA levels compared to early progressors (−1 vs 1.97 median log hGE/mL; $p = 5 \times 10^{-4}$, Fig. 5B). Similar results were seen between D60 and 90. Moreover, ctDNA levels at any time points after D28 were significantly associated with relative TTP from that particular time point (Fig. 5C), and better correlated compared to M-spike or dFLC (Fig. S11A–B), suggesting a prognostic value of ctDNA monitoring after BCMA-CAR.

**Clinical MRD assessment correlates with ctDNA levels**

We assessed whether ctDNA levels correlate with clinical BM MRD measurements. Among our CAR cohort, 28 patients had clinical NGS performed to identify dominant clonal immunoglobulin sequences (ClonoSeq) prior to CAR-T therapy, including seven patients who already had identified dominant clones during prior lines of therapy. Thirteen out of 28 (46%) failed clonal identification on initial assessment; 4 of these patients had clonal sequences identified after assessment of additional archival samples. Overall, the 19/28 patients (68%) with dominant sequences identified by ClonoSeq had a median bone marrow plasma cells (BMPCs) of 52.5% (5–95%) whereas those who failed identification (9/28) had a median BMPCs of 0% (0-5%). This highlights a challenge in identifying dominant sequences in relapsed/refractory patients from BM, as these patients have lower BMPCs (median BMPC 21% in NDMM vs 5% in RRMM in the study cohort). In the remaining patients, MRD assessment was done by next-generation flow cytometry when feasible (NGF, $n = 4$, Supplemental Methods). In contrast, as discussed above, our cfDNA-based assay identified trackable variants in 31/36 (86%) cases to allow disease monitoring.

Overall, 16 patients had paired ctDNA-MRD and BM-MRD (NGS, $n = 12$, NGF, $n = 4$) to allow direct comparison between the two anatomical compartments. NGS-based BM-MRD levels significantly correlated with ctDNA-MRD (rho = 0.92, $p = 0.00003$; Fig. 5D). ctDNA-MRD negativity on D90 was significantly associated with TTP by Kaplan–Meier analysis ($p = 0.0008$, Fig. 5E). Kaplan–Meier analysis of clinical BM MRD vs TTP was not statistically significant, although this analysis is limited by cohort size (Fig. S11C). When patients were categorized by BM MRD and ctDNA, TTP was significantly different among ClonoSeq + /ctDNA + vs ClonoSeq − /ctDNA− vs NGF− /ctDNA+ groups (Fig. S11D).

**CAR-cfDNA levels correlate with flow cytometry enumeration**

In addition to tracking tumor DNA, we designed our assay to detect and track CAR-derived cfDNA molecules, similar to prior work[35] (Supplemental Methods). We compared CAR-cfDNA levels with flow cytometry (FC) enumeration at matched time points. FC and cfDNA quantification of CAR T-cell levels significantly correlated (rho = 0.73, $p = 2.2e\text{-}16$; Fig. 5F). Interestingly, levels of CAR detected in the BM cells and plasma cfDNA were also significantly correlated (Fig. S12A).

The median time to reach a peak level of CAR-cfDNA was 14 days for both ide-cel and cilia-cel, with a median peak concentration of 2.57 (log hGE/mL) for ide-cel, and 2.18 (log hGE/mL) for cilia-cel (Fig. S12B, C). CAR-cfDNA levels and ctDNA burden were inversely correlated (rho = −0.26, $p = 0.0024$ Fig. S12D), demonstrating the relationship between CAR expansion and response.

**Clonal selection detected by ctDNA upon relapse after CAR T-cell therapy**

In addition to providing MRD detection from ctDNA, our approach allows assessment of clonal selection and emergence of genetic alterations. At the time of relapse after BCMA-CAR, we observed the emergence of new alterations in ctDNA, including both SNVs and CNAs. However, the emergence of new CNAs was more recurrent. In particular, we did not discover any emergent SNVs or frameshift mutations in *TNFRSF17*/BCMA (i.e., the target antigen of the CAR T-cell) associated with therapeutic resistance. In contrast, 4/23 patients (17%) had an emergent copy number loss of *TNFRSF17* (Fig. S13A). One of the cases (CAR-10) is depicted in Fig. 6A–C, which had an initial response to ide-cel with a concurrent decrease in ctDNA level and a clinical VGPR. This patient then experienced disease progression on D120 (Fig. 6A). Notably, ctDNA remained detectable and started to rise on D90, which was 30 days prior to clinical relapse, with concurrent copy number loss of *TNFRSF17* (Fig. 6A, B). Moreover, a second focal deletion of *TNFRSF17* was observed in the targeted sequencing data (Fig. 6B insert). This biallelic loss of *TNFRSF17* was confirmed in WGS and CAPP-Seq of the pre-treatment and relapsed BM aspirate, respectively (Fig. S13B, C). Furthermore, the loss of BCMA expression by tumor cells was confirmed by immunohistochemistry (IHC) of the relapsed BM specimen (Fig. 6C). There was another case (CAR-13), who also developed focal loss of *TNFRSF17* (Fig. S14). This suggests that ctDNA may be usable in detecting clonal selection that leads to relapse earlier than current clinical biomarkers, as well as detecting focal genomic loss of the target antigen.

Across all patients with pretreatment and relapse samples available, we observed recurrent clonal selection of multiple genes upon relapse, including deletion of *TNFRSF17* ($n = 4$), amplification of *MYC* ($n = 4$), deletion of *EGR1* ($n = 4$), deletion of *AVL9* ($n = 3$), and deletion of *KRAS* ($n = 5$) (Fig. 6D).

In addition to assessing the emergence and clonal selection of alterations, we assessed the prognostic significance of baseline CNAs and SNVs for outcomes after BCMA-CAR. Five of 36 patients (14%) had underlying copy number loss of *TNFRSF17* identified prior to CAR infusion; 4 of these cases had a monoallelic loss, while only one had a

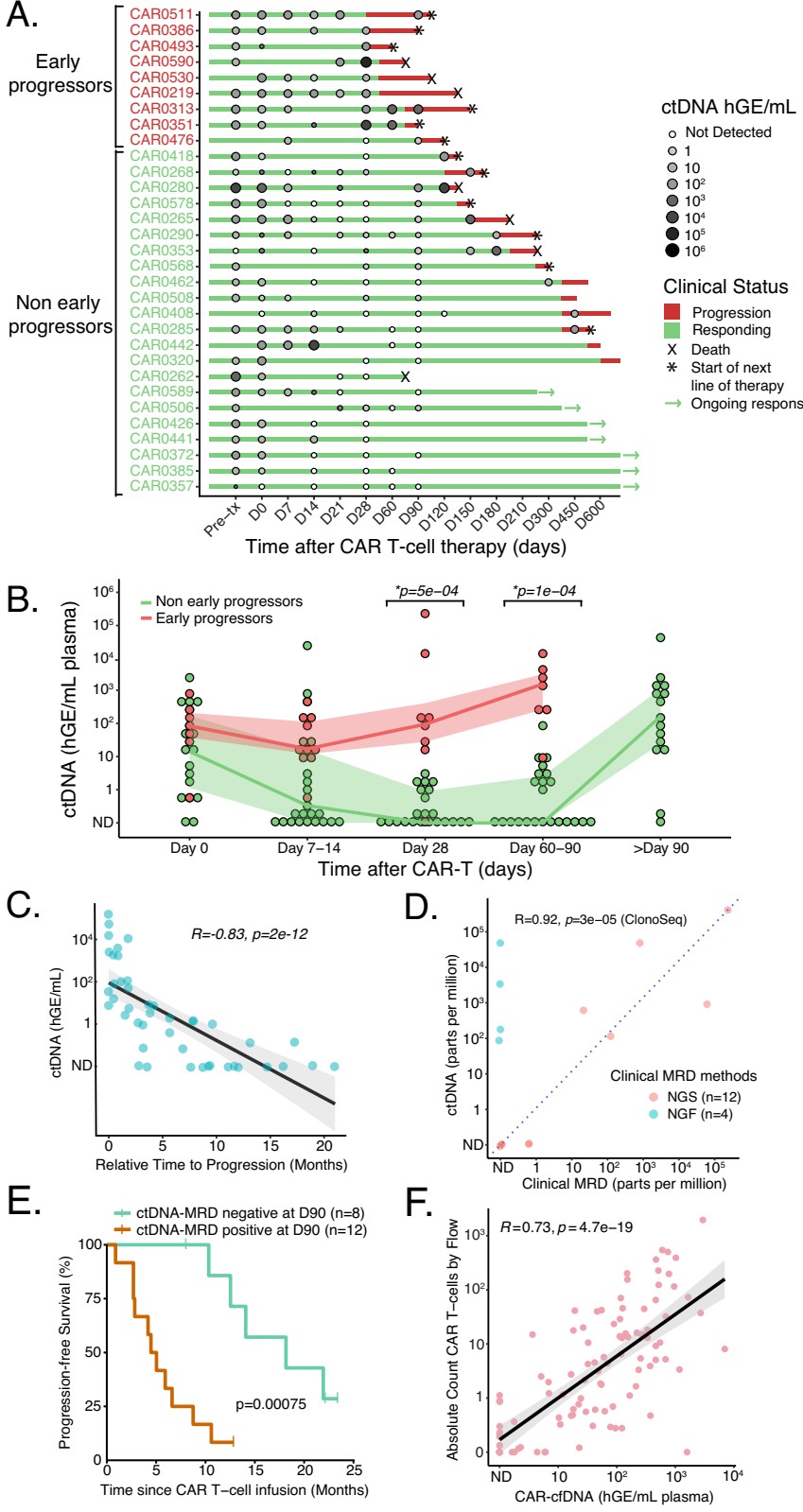

biallelic loss. Notably, there was a discordance between ctDNA assessment and bone marrow assessment, likely representing spatial heterogeneity (Table S5). Interestingly, these patients had significantly shorter TTP after treatment with BCMA-CAR T-cells compared to those who had normal copy numbers prior to BCMA-CAR therapy (*p* = 0.0073 Fig. 6E).

## Levels of ctDNA and molecular features are prognostic for outcomes after BCMA-directed CAR-T therapy

Having applied this multi-faceted approach to cfDNA after BCMA-CAR, we then applied univariate Cox proportional hazards regression models to assess the impact of key patient and disease features on TTP. As expected, higher ctDNA levels on D28 and D90 were prognostic for

**Fig. 5 | Application of ctDNA and CAR-cfDNA monitoring to BCMA-CAR therapy. A** A Swimmer plot of the individual patients receiving BCMA-CAR therapy and their follow-up, overlaid by ctDNA levels. Early progressors; patients who progressed within 3 months following BCMA-CAR. Non-early progressors; patients who did not progress within 3 months. **B** Dynamics of ctDNA after CAR T-cell infusion. Lines indicate the median and interquartile range. *p*-values on day 28 and day 60−90 were calculated by a two-sided Wilcoxon rank-sum test. **C** A scatter plot showing a correlation between ctDNA levels on ≥28 days post-CAR and relative TTP from individual time points. Data are presented as linear regression with 95% CI. The *p*-value was calculated using Spearman's rank-order test. **D** Correlation of

clinical bone marrow MRD and ctDNA at matched time points. Clinical bone marrow MRD was performed using NGS (ClonoSeq, *n* = 12) or NGF (*n* = 4). *p*-Value was calculated using a two-tailed Sperman's rank-order test on samples that had the ClonoSeq test (*n* = 12). The line indicates *y* = *x*. **E** Kaplan−Meier analysis of progression-free survival in patients stratified by ctDNA-MRD positivity on day 90. The *p*-value was calculated using the log-rank test. **F** Correlation of CAR-cfDNA with an absolute count of CAR T-cells by flow cytometry at matched time points. Data are presented as linear regression with 95% CI. The *p*-value was calculated by two-tailed Spearman's rank-order test.

inferior TTP (D28, HR = 1.9, *p* = 2.5 × 10⁻⁴; D90, HR = 2.7, $p = 7.5 \times 10^{-4}$) (Fig. 6F).

Finally, we assessed the prognostic significance of specific genetic alterations genome-wide prior to treatment on outcomes following BCMA-CAR. Interestingly, there were no SNVs that predicted TTP (Fig. S15A). In contrast, CNAs in multiple genes were associated with adverse outcomes to BCMA-CAR. These included amplification of *BRAF* and *MYC*, as well as deletion of *FAM46C* and *CDKN2C*, which are well-described oncogenes and tumor suppressors, respectively, that are implicated in MM pathology (Fig. S15B, C). Indeed, both this pre-treatment CNA analysis, as well as clonal selection analysis implicate *MYC* as a key determinant of resistance to MM immunotherapy. These findings suggest that our tool can assess ctDNA quantity and CNAs simultaneously, and has utility for predicting clinical outcomes after BCMA-CAR therapy.

## Discussion

While BM-based MRD is well established in MM, minimally invasive approaches to disease detection and characterization remain an unmet need in MM. Prior studies have demonstrated the feasibility and utility of liquid biopsy in MM via ctDNA detection. However, these early approaches have had challenges in the detection of low-level MRD due to limited sensitivity[15,16,18,19,43–45].

In this study, we designed a comprehensive targeted gene panel to characterize and track ctDNA via peripheral blood in patients with MM. By performing CAPP-Seq, we show that an abundance of tumor markers (i.e., a median of over 80 SNVs per case), are detected using this panel for disease tracking, and ctDNA level was associated with disease burden and adverse risk factors in MM. Notably, ctDNA levels were comparable in secretory vs oligo/non-secretory patients, making ctDNA a potential blood biomarker in patients who have lost their circulating protein-based biomarker. In addition, we demonstrate that changes in ctDNA levels after 1 month of therapy corresponded with the best clinical response following both cellular and non-cellular therapies.

Recently, novel immunotherapies including anti-BCMA CAR T-cell therapy have emerged as promising entities in treating RRMM. However, there are no established predictive biomarkers for treatment response or duration of response after CAR-T. Our study demonstrates that ctDNA tracks with disease burden and detects MRD at a comparable sensitivity to conventional BM-based MRD, which requires an invasive procedure. Additionally, ctDNA levels at any given time after 28 days following BCMA-CAR are associated with TTP from that time point. By incorporating oligonucleotides targeted to CAR vectors, we demonstrate simultaneous detection of CAR-cfDNA and ctDNA, enabling tracking of the expansion and persistence of CAR T-cells along with therapeutic response. We show that peak CAR-cfDNA levels are associated with TTP in our ide-cel cohort.

Another application of our assay is to explore the genomic mechanisms of resistance to therapy. Resistance to CAR T-cell therapy is thought to be multifactorial, including tumor intrinsic factors, CAR T-cell fitness, and tumor microenvironment factors[46]. Emergent BCMA loss with or without BCMA mutation is known to be one of

these mechanisms of relapse after anti-BCMA therapy[9,47]. It is also reported in 6−7% of ND or RRMM patients, who never had an exposure to BCMA-targeted therapy[48,49]. Our assay detected clonal selection after BCMA-CAR, where a clone harboring loss of BCMA was selected and became dominant, with a concurrent rise in ctDNA. This ctDNA rise and clonal selection were observed prior to biochemical relapse as assessed by clinical myeloma laboratory markers, suggesting that ctDNA may be useful for early disease relapse detection, as well as revealing resistance mechanisms. In our cohort, we observed that 5/36 patients (14%) had underlying BCMA loss prior to BCMA-CAR, 1 of whom later had a clonal evolution with a focal BCMA loss resulting in loss of both alleles. Additionally, three patients acquired emergent BCMA loss upon relapse. Those who had underlying BCMA loss had shorter TTP compared to those who did not. We note that these patients were also enriched for underlying amplification of *MYC*, which is implicated in conferring poor prognosis[50], and reported as an emergent CNV upon relapse after CAR-T[47]. In our cohort, we also noted the selection of clones harboring amplification of *MYC*, or deletion of *EGR1*, *AVL9*, or *KRAS* upon relapse after BCMA-CAR. Future studies will be needed to determine if these alterations in genes outside of *TNFRSF17* (BCMA) represent specific mechanisms of resistance, or are simply markers of more aggressive disease biology.

While there are multiple uses of ctDNA-based assessment in MM as outlined above, it is important to recognize the limitations of this assay. While we have demonstrated the ability to detect both SNVs and CNAs using this approach, the sensitivity for CNA detection is lower, as described in previous ctDNA approaches[40]. Furthermore, we did not explore the detection of translocations using our approach, which is a key recurrent type of alteration in MM. Indeed, we envision ctDNA assays will supplement—rather than replace—current bone marrow assays, which will remain important to establish a pathological diagnosis and to assess structural alterations such as CNAs and translocations using gold-standard FISH. Indeed, approaches such as this are likely to find initial use in research and discovery settings—such as the exploration of mechanisms of response and resistance to therapy—prior to gaining use in the clinic. The strengths of our approach are to non-invasively characterize MM, to monitor residual disease, and to assess resistance mechanisms to therapy as patients undergo treatment. Additional limitations of our study include a small cohort size from a single center and a short follow-up. Further investigation in a larger cohort will allow further analysis such as rigorous subgroup analysis of patients undergoing ide-cel vs cilta-cel. A larger cohort may also provide deeper insights via longitudinal therapeutic monitoring and may reveal additional genomic mechanisms of resistance. Future studies validating pre-treatment BCMA loss and association with clinical outcomes, which has significant clinical implications, will be needed.

Taken together, we demonstrate that a non-invasive ctDNA assay can inform multifaceted genomic information that is useful for genotyping, disease monitoring, early detection of disease relapse, and detecting mechanisms of treatment resistance. Overall, this study indicates a strong potential to improve the clinical management of MM using personalized ctDNA analysis.

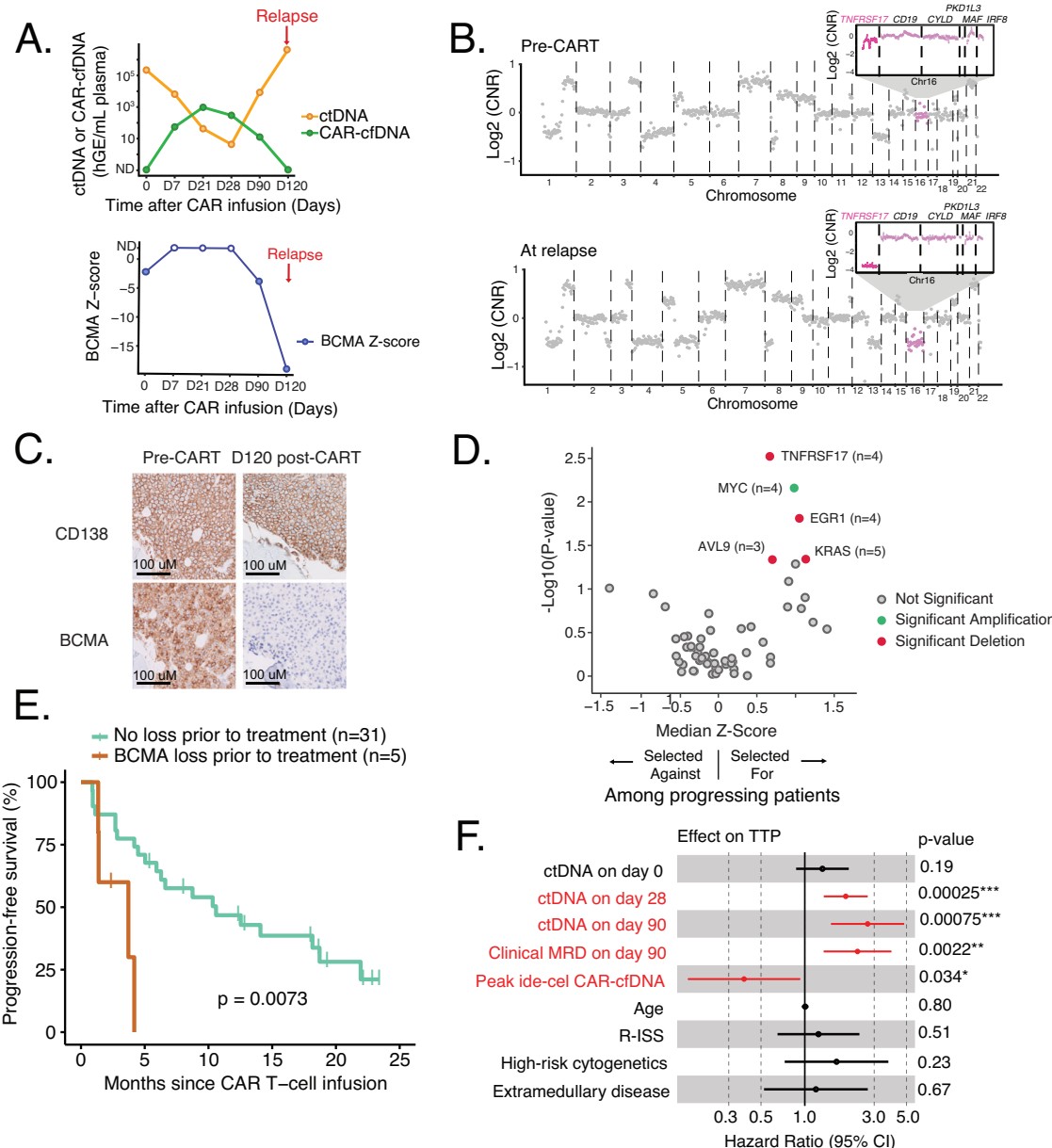

**Fig. 6 | Genomic determinants of resistance to BCMA-CAR therapy. A** A case of early detection of relapse by ctDNA, showing the dynamics of ctDNA and CAR-cfDNA (upper panel), as well as the dynamics of *TNFSRSF17* (BCMA) z-score (lower panel). Open points reflect non-significant *q*-score indicating the BCMA deletion was not detectable. **B** Genome-wide copy number profile inferred from cfDNA WGS of the patient depicted in A). Emergent chromosome 16 deletion is magnified to show further focal deletion in *TNFRSF17* (BCMA), which was detected by CAPP-Seq. Segmentation was performed using the DNAcopy R package. CNR, copy number ratio. **C** Immunohistochemistry of the bone marrow core of the patient depicted in **A** prior to BCMA-CAR (left) and upon relapse (right) demonstrating loss of BCMA.

**D** The volcano plot shows a clonal selection of somatic copy number alterations at individual gene levels in patients who relapsed (*n* = 23). Positive *z*-score reflects clonal selection in patients at relapse. *p*-values were calculated using a two-sided Wilcoxon rank-sum test. Copy number alterations that were significantly clonally selected (*p* < 0.05) are highlighted in green for amplifications, and red for deletions. **E** Kaplan–Meier analysis of progression-free survival in patients with and without underlying copy number loss of BCMA prior to treatment. **F** Forest plot of variable effects on time to progression (*n* = 36). The plot depicts hazard ratios ± SD calculated by a univariate Cox proportional hazards regressions model, with significant values shown in red (*p* < 0.05). Error bars reflect a 95% confidence interval.

## Methods

### Patients

Patients with MM or other plasma cell disorders who were treated under standard-of-care at Stanford Cancer Center were enrolled in this study between February 2017 to July 2024. A total of 64 patients had samples collected and were included in this study. Seventeen patients had newly diagnosed and/or treatment-naïve disease, and 47 patients had relapsed/refractory disease. Thirty-six patients subsequently underwent CAR T-cell therapy under the standard of care (idecabtagene vicleucel, ide-cel; *n* = 30; ciltacabtagene autoleucel, cilta-cel;

*n* = 6). The study was approved by the Stanford Institutional Review Board (protocol #43375) in accordance with the Declaration of Helsinki, and all patients provided written informed consent. There was no compensation to the participants. Additional information can be found in the Supplemental Methods.

### Sample collection and processing

Peripheral blood samples were collected in Cell-Free DNA Collection Tubes (Roche, Pleasanton, CA) or Cell-Free DNA Blood Collection Tube (BCT) (Streck, La Vista, NE) and processed within 7 days by

centrifugation at 1800 g for 10 min. Plasma and plasma-depleted whole blood were subsequently isolated and cryopreserved at −70 °C as previously described[34]. Additional information can be found in the Supplemental Methods.

## Designing of a targeted gene panel for multiple myeloma

In order to detect genomic alterations in MM, a custom 570 kb oligonucleotide panel (IDT, Coralville, IA) was designed using publicly available whole genome sequencing (WGS) and WES data[26–29]. This panel includes genomic regions known to be recurrently mutated or of known biological significance in MM. In addition to the MM panel, supplemental custom oligonucleotide panels were designed to target ide-cel and cilia-cel transgenes, T-cell receptor genes, and genes known to be implicated in immune evasion. Coding genes captured in this panel are shown in Table S1. We define coding mutations as non-synonymous gene mutations, whereas non-coding mutations are used to refer to both synonymous gene mutations and mutations in non-coding or intergenic regions.

## Statistical analysis

Statistical analyses were performed using R (v.4.0.3), MATLAB (R2020b), and GraphPadPrism (v.9.0.0). Analyses for significant differences between the two groups were assessed using the Wilcoxon rank-sum test. Survival probabilities were estimated using the Kaplan–Meier method, and the survival of the two groups was compared using a log-rank test. We considered time to progression (TTP) wherein an event was defined as disease progression or relapse as the survival endpoint. ctDNA detection was determined with our previously described Monte-Carlo-based ctDNA detection index[25]. The statistical significance of Pearson or Spearman correlation was determined using $t$ statistics. Regression analysis of covariates was conducted by Cox proportional hazards modeling, with $p$-values assessed using the log-likelihood test. For Cox regression analyses, continuous variables were used as input and standardized using Fisher's $z$-score transformation. Units were defined as follows: ctDNA, log10(haploid genome equivalents per milliliter; hGE/mL), CAR-cfDNA, log10(genomes/mL).

## Reporting summary

Further information on research design is available in the Nature Portfolio Reporting Summary linked to this article.

# Data availability

The genotyping data generated in this study have been deposited in Figshare (https://doi.org/10.6084/m9.figshare.24803172). Due to patient confidentiality and restrictions from patient consent forms, the raw genomic data are protected and are not available per the institutional review board. The processed genomic data generated in this study are provided in the Supplementary Information/Source Data file. For additional inquiries, please contact the corresponding authors at dkurtz@stanford.edu. Source data are provided with this paper.

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

## Acknowledgements

This study is supported by the National Cancer Institute (K08CA241076 to D.M.K.), the American Cancer Society-Stanford Cancer Institute IRG Pilot Grant (D.M.K.), the Gabrielle's Angel Cancer Research Foundation Medical Research Award (D.M.K), the Leukemia Research Foundation New Investigator Research Grant (D.M.K.), and the Shanahan Family Foundations (D.M.K.), and CKD Family Fund (M.L.). H.H. is supported by the SITC-Merck Cancer Immunotherapy Clinical Fellowship. S.S. was supported by Stanford Clinical and Translational Science KL2 Career Development Award program (KL2 TR003143), Stanford Cancer Institute/American Cancer Society Pilot Grant 2022, and Doris Duke Charitable Foundation.

## Author contributions

H.H. and D.M.K. designed the study; H.H., D.I., and S.S. enrolled patients; K.T., M.C., and H.H. processed samples; H.H. and M.C. performed experiments; H.H., V.H., and S.S. collected clinical data; H.H., M.C., and A.M.M., and D.M.K. analyzed the data; B.Sa. designed CAR T-cell flow cytometry and S.Sy. conducted flow cytometry; O.S. performed IHC; H.S. provided Heme-STAMP sequencing data; H.H., M.C., and D.M.K. wrote the manuscript. B.S., G.D., T.Z., M.L., S.A., D.I., D.M., and M.S.K. critically reviewed the manuscript.

## Competing interests

B.J.S.: consultancy: Foresight Diagnostics. M.L.: advisory board/consulting for Janssen, BMS, and Kite. D.M.: patent: Pharmacyclics supporting ibrutinib for chronic graft-vs-host disease. Research funding and consultancy: Pharmacyclics, Kite Pharma, Adaptive Biotechnologies, Novartis, Juno Therapeutics, Celgene, Janssen Pharmaceuticals, Roche, Genentech, Precision Bioscience, Allogene and Miltenyi Biotec. M.S.K: research funding: Nutcracker Therapeutics, CRISPR Therapeutics; Consultancy: Daiichi Sankyo, Myeloid Therapeutics. S.S.: research funding: Magenta Therapeutics, BMS, Allogene, Janssen; Consultancy: Magenta Therapeutics, BMS, Janssen, Sanofi, Oncopeptides, Takeda. D.M.K.: patents: D.M.K. reports issued patents pertaining to ctDNA-MRD detection assigned to Stanford University and licensed to Foresight Diagnostics. Consultancy: Foresight Diagnostics, Roche, Genentech, and Adaptive Biotechnologies. Stock and Equity Ownership: D.M.K. reports equity ownership in Foresight Diagnostics. The remaining authors declare no competing interests.
