## [Transparent Peer Review file · Nature Communications]

Deciphering response dynamics and treatment resistance from circulating tumor DNA after CAR T-cells in multiple myeloma

Corresponding Author: Dr David Kurtz

Version 0:

Reviewer comments:

Reviewer #1

(Remarks to the Author)

The authors developed a robust and ultra-sensitive CAPP-Seq (Cancer Personalized Profiling by Deep Sequencing) hybrid capture approach for ctDNA in MM. The sensitivity of ctDNA detection increased by analyzing not only exome DNA but also non-coding DNA. The authors serially measured ctDNA in MM patients treated with BCMA CAR-T cells, providing valuable and novel data. Unlike solid tumors, in the case of MM, MRD can be detected by analyzing BM cells, although collecting BM samples is minimally invasive. Therefore, the critical question is whether the sensitivity of MRD detection by ctDNA measurement is higher, comparable, or lower than that of MRD detection by BM analysis using NGS or flow cytometry.

Major

1. In Fig. 5A, the authors are recommended to compare the ability of MRD detection between ctDNA and other methods.
2. Since MM sometimes progresses locally or in patchy regions, ctDNA may be better at detecting MRD or acquired genetic alterations than BM analysis in some cases. Can the authors present such cases?

Minor

1. The legend for Fig. 2A needs to be revised. Does the lower panel show DLBCL cohorts?

Reviewer #2

(Remarks to the Author)

Hosoya and colleagues report the use of ctDNA approach for characterizing tumor genomics, assessing treatment responses, and detecting early relapses in patients with multiple myeloma. They use this approach in 2 cohorts of patients- 1) 56 patients with newly diagnosed or relapsed/refractory disease and 2) 28 patients with RRMM treated with BCMA CART cell therapy. Overall well done study with important findings, however limited by the relatively small sample size.

The study is well done and the manuscript is clearly written.

Few comments/suggestions:

1. Lines 308: Please confirm and clarify that the BCMA loss noted in the 5 patients prior to treatment with BCMA CAR T cells was monoallelic
2. Did these 5 patients have other high-risk features like del 17p?
3. I would suggest table 1 be broken down as: 1. Newly Diagnosed; 2) RRMM (non-cellular therapy) and RRMM-BCMA CART if possible. Also please indicate what cytogenetic changes are included as 'high risk' as a footnote.
4. Does Fig 3E list all of genes assessed? I did not see TNFSRSF17 (sorry if i missed)?

Reviewer #3

(Remarks to the Author)

Hosoya et al. present a study analyzing the use of ctDNA in MM patients treated with a range of therapies, including CAR-T.

The main idea of the manuscript is that ctDNA is a less invasive method and comparable to WES/WGS/FISH in BM aspirates, and serves as prognostic/predictive biomarker.

Despite the interesting premise, the article needs revision in order to achieve more rigorous language, and the study itself needs more adequate tests/validations in order to support claims made.

Here are a few of these points, although this list is not all inclusive:

- 1-Figure 2A does not match its caption. There is mention of WGS, but caption says DLBCL? Where are the driver/most common MM genes?
- 2-Figures 2B,C the comparisons do not make sense. Why compare number of SNVs between different cancers?
- 3-Figure 2D, how many of these mutations in teal are not part of the STAMP panel? How do authors know these mutations are real (ground truth)? How to know the overlap of mutations detected in both assays?
- 4-Figure 2E-what is unit of limit of detection? In text authors mention coding/noncoding mutations, does it mean coding and non-protein coding genes, or synonymous vs. non-synonymous mutations?
- 5-Figure 3A: How many of which? How many overlapped (same patient same day)? What is genotyping? WGS?
- 6-Statement on line 145, regarding Figure 3D: What is the rationale for this statement? Is this a general observation or anecdotal? Coding vs. non-coding genes or synonymous vs. non-synonymous mutations?
- 7-Please provide a version of 3E for conventional WES/WGS from marrow for comparison purposes.
- 8-Line 158, re: Fig S1, VAFs seem to be biased towards detection in BM but not in plasma. Meaning that there is a lower detection from ctDNA, in many cases, of all genes. Venn diagram dimensions are misleading (40% of area of BM detected SNVs should be outside the intersection). How are the ones only detected in ctDNA considered real or artifact?
- 9-Line 160, re: Fig S2A: These comparisons seem to be to in overall number of mutations in the cohort, instead of determining agreement between two assays in each patient. Please address this.
- 10-Line 162: What does this sentence mean?
- 11-Line 165: Or maybe they are artifacts? Or derived from other cancers these patients may have? Is there a ground truth to compare to?
- 12-Line 166: Because they are more frequent and more clonal? Most mutations in MM are low frequency, apart from these, and given the limited cohort size of this study, it is expected that only these mutations are found in a reasonable number.
- 13-Line 176, re: Fig S3A: Unfortunately these CNA results are not very convincing nor impressive. It seems it cannot replace regular FISH, given does not do translocations, thus BM aspirate remains necessary.
- 14-Section starting on line 206: ctDNA collected 28 days after therapy initiation (please provide details on treatment of non-CAR-T patients) is hardly a predictive biomarker, unless the ctDNA were collected before initiation of therapy. Please compare fold change of M-Spike/SFLC to determine that ctDNA is superior.

Reviewer #4

(Remarks to the Author)

In this study, the authors have developed a circulating tumor DNA approach for the characterization of tumor genomics, monitoring of treatment responses, and early detection of relapses in multiple myeloma. Through the analysis of 352 specimens from 56 patients, both newly diagnosed and those with relapsed/refractory disease, the study establishes not only a correlation between ctDNA and patient outcomes but also extends its application to the tracking of CAR-specific cell-free DNA in patients undergoing anti-BCMA CAR T-cell therapy. Notably, the inverse correlation between ctDNA levels and the relative time to progression post-BCMA-CAR therapy, along with the concordance of ctDNA-MRD with clinical bone marrow MRD, are key findings. Additionally, the capability of ctDNA-MRD to predict clinical relapse and identify resistant clones marks a notable advancement.

However, some areas of the study warrant further clarification.

In the study, the authors note that overall, in samples where the ctDNA Allele Frequency (AF) was greater than 0.1%, the sensitivity for detecting bone marrow-derived single nucleotide variants (SNVs) was 82%. However, the study does not explicitly address the rate of false positives associated with this AF level. Clarification on the false positive rate in this context would be a valuable addition.

Furthermore, the methodology section of the paper indicates that an AF greater than 0.3% was used for analysis. This higher threshold potentially increases the likelihood of false negatives, which in turn might lead to an underestimation of the ctDNA positive rate. Such a discrepancy raises concerns about the accuracy of ctDNA detection and the criteria used to define ctDNA positivity. It would be beneficial for the authors to address this issue, providing insights into their rationale for choosing these specific thresholds and how they impact the study's findings.

The Kaplan-Meier analysis of progression-free survival, stratified by clinical MRD on day 90, offers valuable insights but also raises questions about the biological underpinnings of this particular time frame's effectiveness. Providing a biological rationale for this observation would enrich the study's depth.

The authors of the study make a comparison between their method and earlier methods referenced in sources 16 to 22. However, this comparison is quite brief. A more detailed analysis highlighting how their method improves upon or differs from these previous methods would help readers better understand the progress made in this study.

The use of a specially designed target bed file, specifically for multiple myeloma (MM), is an important aspect of this study. In Table S1 of the study, the authors list the genes they are focusing on, but they don't provide much information about other significant genomic areas. Providing the full details of the target bed file or the bait sequences used in the study would make the findings more reliable and complete.

Considering the methodological and conclusion similarities with a study published in Cancer Cell in 2023 (reference 34), a more detailed discussion on the novelty of the current study and its distinct contributions compared to the prior research is recommended.

Correcting the truncated title in reference 37 is essential to ensure the accuracy and completeness of the study's citations.

Version 1:

Reviewer comments:

Reviewer #1

(Remarks to the Author)

The authors have successfully addressed all of my concerns.

Reviewer #2

(Remarks to the Author)

The authors have adequately addressed the reviewer comments.

Reviewer #3

(Remarks to the Author)

Missing points originally raised but not addressed by authors:

"1-Figure 2A Where are the driver/most common MM genes?"

"3-Figure 2D, how many of these mutations in teal are not part of the STAMP panel? How do authors know these mutations are real (ground truth)? How to know the overlap of mutations detected in both assays?" This remains an issue as the authors cannot claim their assay is better because it calls more mutations, since these could be false positives. A correct approach would be to sequence both tumor and ctDNA with both assays.

"7-Please provide a version of 3E for conventional WES/WGS from marrow for comparison purposes." A true validation of false positives/false negatives is to base on a ground truth. I agree that the goal is not to replace BM-WES, but in order to determine how many new mutations/clones exist/disappear in sequential assays, it is important to determine how accurate assay is. However, the authors are the ones who claimed that their assay in most MM cases was comparable to invasive tumor genotyping, and now have changed their text to "... suggests that in most cases of MM, where the ctDNA 187 amount is above this level, BM-derived SNVs can reliably detected from the plasma".

"11-Line 165: Or maybe they are artifacts? Or derived from other cancers these patients may have? Is there a ground truth to compare to?" Once again, the question is not regarding bioinformatics analysis, but false positives that can emerge from the assay itself. We need a control/groundtruth using another assay (WES/WGS, etc.) for a true validation.

"13-Line 176, re:" Fig S3A: Unfortunately these CNA results are not very convincing nor impressive. It seems it cannot replace regular FISH, given does not do translocations, thus BM aspirate remains necessary." I feel the authors are trying to associate as many features the to product as then can, when instead they should focus on providing strong evidence of its main use and reliability. This is another case where even regular WES/WGS has difficulty in assessing CNVs, and ctDNA with an even more limited gene set, makes it even more challenging. I stand by my original comments.

"14-Section starting on line 206: ctDNA collected 28 days after therapy initiation (please provide details on treatment of non-CAR-T patients) is hardly a predictive biomarker, unless the ctDNA were collected before initiation of therapy. Please compare fold change of M-Spike/SFLC to determine that ctDNA is superior." Same as above. This claim should not be part of the article.

Reviewer #4

(Remarks to the Author)

The authors have provided clear and thorough responses to the questions raised, addressing all concerns. I have no further questions, and I believe their detailed answers have contributed to enhancing the overall quality of the manuscript.

Version 2:

Reviewer comments:

Reviewer #3

(Remarks to the Author)

The authors have addressed the questions this reviewer raised, which mainly were requests to focus the articles claims on the usability of the assay, taking into consideration both its strengths and limitations.

Such novel technologies are very important for non-invasive observation of MM patients, but unfortunately cannot yet replace invasive procedures for the thorough characterization of the tumor burden. Hopefully soon.

I believe this article is a great contribution to the field.

Response to Reviewers

Deciphering response dynamics and treatment resistance from circulating tumor DNA after CAR T-cells in multiple myeloma (NCOMMS-24-01077A)

We thank the reviewers for their comments as well as their insightful critiques and suggestions, which helped us prepare a revised manuscript that we feel is significantly improved. Major changes based on reviewers' suggestions include:

- 1) An expansion of our CAR T-cell cohort, with addition of 8 new CAR-T patients bringing the total cohort size to 36 patients. The demographic tables (Table 1, Table S2) were updated to incorporate these additions. These additions serve to further validate the findings of this study.
- 2) Further comparison of conventional bone marrow-based MRD detection and our ctDNA-based method. This is augmented by the inclusion of the additional patients above. This comparison highlights multiple challenges with current bone-marrow based MRD methods, including 1) identifying tumor-specific DNA sequences for NGS-based bone marrow MRD, and 2) suboptimal sensitivity with current approaches. These comparisons can be found in Figure R1 (Figure 5D in the updated manuscript), as well as outlined below.
- 3) An expansion of healthy controls in order to apply more robust false positive rate of ctDNA detection. We added 8 new healthy controls bringing the total control panel size to 26 patients.
- 4) Further dedicated analysis of the analytical performance of our assay, including description of the rate of false positives, as well as allele frequency thresholds for detection of alterations. We have also expanded on the description of the assay in the methods, including specific descriptions of the thresholds for detecting variants and MRD. These methodologies can be found in page 3 of supplemental manuscript.

In the following document we provide detailed responses to each of the reviewers' comments and concerns.

Reviewer #1:

The authors developed a robust and ultra-sensitive CAPP-Seq (Cancer Personalized Profiling by Deep Sequencing) hybrid capture approach for ctDNA in MM. The sensitivity of ctDNA detection increased by analyzing not only exome DNA but also non-coding DNA. The authors serially measured ctDNA in MM patients treated with BCMA CAR-T cells, providing valuable and novel data. Unlike solid tumors, in the case of MM, MRD can be detected by analyzing BM cells, although collecting BM samples is minimally invasive. Therefore, the critical question is whether the sensitivity of MRD detection by ctDNA measurement is higher, comparable, or lower than that of MRD detection by BM analysis using NGS or flow cytometry.

We appreciate the reviewer's recognition of the value of our study and for their detailed comments. We also agree with the reviewer's comment putting our study in the context of the current clinical paradigms in MM, where bone marrow based MRD assessments are currently utilized. We have significantly expanded the comparison between our blood plasma-based MRD method with traditional BM based MRD assessments, as outlined below in Q1. This includes additional analyses comparing the sensitivity for detection of residual disease by each method, as well as comparison between methods to identify clones for tracking MRD.

In addition, to this technical comparison below, we also would like to highlight the clinical challenges facing traditional BM-based MRD, as well as the improvements that blood plasma based MRD can afford. Specifically, bone marrow biopsies are significantly more invasive and painful than blood-based assessment, and patients are frequently hesitant to undergo this procedure; therefore, blood-based assessments would represent a significant step forward for MRD in MM. Blood based assessment can also allow more frequent assessment of MRD, as sustaining MRD negativity is more valuable than achieving MRD negativity, especially in context of MRD guided therapy. Blood based assessment through CAPP-Seq can also account for tumor DNA from extramedullary sites, a current limitation of marrow based MRD testing. We have added this context to the discussion of the manuscript.

Major

1. In Fig. 5A, the authors are recommended to compare the ability of MRD detection between ctDNA and other methods.

We thank the reviewer for inviting further comparison of MRD detection between standard methods and ctDNA. Paired clinical MRD and ctDNA data was available for a total of 16 patients post-CART therapy. The methods for clinical bone marrow MRD assessment used were: NGS (ClonoSEQ; n=12) and next-generation flow cytometry (NGF; n=4). NGF was only used when a baseline pre-CAR-T clone could not be identified for NGS based testing on pre-treatment bone marrow, a limitation of the NGS based method for MRD detection. MRD quantification by ClonoSEQ and ctDNA was highly correlated ($R=0.92$, $p=3e-05$) as shown in Fig R1 (Fig 5D in the manuscript). There were four patients where NGF was done for marrow based MRD after failure of ClonoSEQ ID. These 4 cases had bone marrow disease involvement of 0-1%, and NGF all returned negative while ctDNA were detected at greater than 86 per million. Out of these 4 cases, 2 patients were found to have disease progression on the same

date, by blood test (free light chains) or PET/CT scan, highlighting a challenge in relying on bone marrow as a source of tumor.

We showed Kaplan-Meier analysis of patients based on ctDNA-MRD +/- and Bone Marrow MRD +/- in Fig. 5E and Fig. S11C, respectively (pasted below for the reviewer). We performed additional Kaplan Meier analysis in these patients who had i) ClonoSEQ+/ctDNA+ (n=5), ii) ClonoSEQ-/ctDNA- (n=5) and iii) NGF-/ctDNA+ (n=4) and observed that ctDNA was comparable to Bone Marrow based ClonoSEQ in prognostication (Fig R2, Fig. S11D in the manuscript). Specifically, a total of 5/5 patients who were Bone Marrow MRD+/ctDNA+ progressed within 12 months, while 3/4 patients who were Bone Marrow MRD-/ctDNA+ progressed within 12 months. In contrast, only 1/5 patients who were Bone Marrow MRD-/ctDNA- progressed within 12 months. We also observed that in some cases, ctDNA better reflects clinical disease status compared to Bone Marrow MRD (Fig R3). Specifically, CAR-02 and CAR-07 are cases that progressed within 90 days of CAR-T infusion that had negative clinical MRD on day 28 and day 90, respectively. CAR-07 had existing extramedullary disease whereas CAR-02 developed new extramedullary disease upon relapse that was missed by bone marrow assessment. These same timepoints were detectable by ctDNA-MRD, highlighting the utility of ctDNA analysis in the setting of extramedullary disease.

Figure R1 (Fig 5D in manuscript) Correlation between ctDNA and clinical bone marrow ClonoSEQ or next-generation flow cytometry. There was a significant correlation between ctDNA and ClonoSEQ quantification.

Fig. 5E and Fig. S10C (reproduced for the reviewer) Kaplan-Meier analysis of PFS of patients with ctDNA-MRD positivity (A) and BM-MRD positivity (B).

Figure R2 (Fig S11D in manuscript) Kaplan-Meier analysis of PFS of patients with i) ClonoSEQ+/ctDNA+ (n=5), ii) ClonoSEQ-/ctDNA- (n=5) and iii) NGF-/ctDNA+ (n=4). The clinical outcomes were concordant between Bone Marrow ClonoSEQ results and ctDNA results.

Hosoya et al. *Deciphering response dynamics and treatment resistance from circulating tumor DNA after CAR T-cells in multiple myeloma.*

Figure R3 Swimmer plot analysis with cases with paired clinical Bone Marrow MRD and ctDNA-MRD. While BM-MRD and ctDNA-MRD are overall concordant, there are a few important discrepancies including CAR-02 and CAR-07 where ctDNA was positive upon relapse when Bone Marrow MRD was negative.

2. Since MM sometimes progresses locally or in patchy regions, ctDNA may be better at detecting MRD or acquired genetic alterations than BM analysis in some cases. Can the authors present such cases?

The reviewer presents a strong point and rationale for use of ctDNA to overcome spatial heterogeneity of multiple myeloma.

As presented above, current bone marrow-based MRD assays have challenges in cases where there is minimal bone marrow involvement and we cannot identify dominant sequences by ClonoSeq, which occurred in approximately one-third of cases (n=28, Fig R4). In these cases, NGF-based MRD was also not informative. We show that ctDNA genotyping is more successful and we were able to identify SNVs>15 in 86% of the cases in our cohort (n=36, Fig R4). Furthermore, we show that ctDNA-MRD is at least comparable to a current MRD assay although we recognize our cohort size is small (Fig R1).

Figure R4 Bar graph showing success rate of genotyping using ClonoSeq assay vs CAPP-Seq assay for patients undergoing CAR T-cell therapy.

The reviewer also highlights the important point that ctDNA is able to detect acquired genetic alterations.

As an example, we present a case (CAR-06), who saw a reduction of disease burden in the bone marrow on day 28, however ctDNA showed clonal evolution with increasing allele frequencies, particularly those mutations that were $AF < 0.3\%$ in the bone marrow (Fig R5; Fig S3 in the manuscript). This patient relapsed on day 82 with increased light chains. PET/CT showed interval decrease in bone marrow hypermetabolism, however there were some areas with increased metabolism including focal nodal and bone uptake. Both SNVs and CNVs demonstrated remarkably different makeup of somatic alterations in the plasma ctDNA as compared to the bone marrow (Fig R5B-C). This case highlights spatial heterogeneity of multiple myeloma, and underscores the importance of developing an assay that can assess as a whole tumor.

Figure R5 (Fig S3 in manuscript) SNV dynamics of ctDNA vs bone marrow during CAR-T therapy in CAR-06, who relapsed on day 82. (A) Dynamics of each SNV in the plasma and bone marrow. Orange dots are SNVs detected in the BM at VAF > 0.3%, teal dots are SNVs detected only in the plasma at VAF > 0.3%. (B) A scatter plot of allele frequencies of each SNVs in the plasma vs bone marrow on day 28, demonstrating that many mutations were detected at much higher VAF in the blood plasma compared to bone marrow. (C) Copy number plot of plasma vs bone marrow on day 28 across the whole genome, showing that copy number alterations were more readily detectable from the blood plasma.

Minor

1. The legend for Fig. 2A needs to be revised. Does the lower panel show DLBCL cohorts?

We thank the reviewer for their comments here and agree this can be presented more clearly, and agree that the point of this figure was not clear in the initial submission. This panel was included because plasma-based ctDNA and CAPP-Seq are established techniques in non-Hodgkin lymphomas (Scherer et al., *Sci Transl Med.* 2016, PMID 27831904, Kurtz et al., *J Clin Oncol.* 2018, PMID 30125215, Kurtz et al., *Nat Biotechnol.* 2021, PMID 34294911). Targeted NGS approaches for tracking MRD are particularly effective for lymphomas due to the stereotyped occurrence of mutations in focused regions of the genome due to the activity of activation induced deaminase (AICDA). Multiple myeloma has a similar stereotyped distribution of mutations, which therefore makes this an excellent candidate for panel-based targeted NGS for ctDNA-MRD.

However, as this paper is focused on multiple myeloma, we agree that it should not include DLBCL cohort in the main figure, and thus we have moved it to supplemental materials (Figure S1).

Reviewer #2:

Hosoya and colleagues report the use of ctDNA approach for characterizing tumor genomics, assessing treatment responses, and detecting early relapses in patients with multiple myeloma. They use this approach in 2 cohorts of patients- 1) 56 patients with newly diagnosed or relapsed/refractory disease and 2) 28 patients with RRMM treated with BCMA CART cell therapy. Overall well done study with important findings, however limited by the relatively small sample size.

The study is well done and the manuscript is clearly written.

We appreciate the reviewer's comments on the design of our study and the importance of our findings. We also recognize the limitations of a study with a small sample size. Despite this, as our manuscript is the first to robustly characterize the performance of targeted ctDNA-MRD analysis in MM, particularly in the setting of novel immunotherapies such as BCMA CAR T-cells, we believe this manuscript adds significant novelty to the literature. However, we do acknowledge the small sample size in our initial submission, and we have expanded the cohort in our resubmission to address the by adding 8 additional CAR-T patients. The main results of the manuscript have not changed with the inclusion of these patients, and are indeed strengthened by their inclusion. This first publications will provide the framework to expand this technology into a wider population of patients in the future.

Few comments/suggestions:

1. Lines 308: Please confirm and clarify that the BCMA loss noted in the 5 patients prior to treatment with BCMA CAR T cells was monoallelic

We thank the reviewer for this interesting question and the opportunity to further clarify our findings related to pre-treatment BCMA loss. Below table lists the 5 patients who we detected

genomic loss of BCMA prior to CAR T-cell therapy. We also added BCMA expression by bone marrow immunohistochemistry as a reference (Table R1; Table S5 in the manuscript). As our CAPP-Seq assay sequences bulk tumor or cell-free DNA, it does not provide granularity of clonal composition with different BCMA copy numbers amongst clones (i.e., single tumor cells are not resolved). However, we can assess the aggregate inferred copy number state averaged across all malignant cells as described here. The aggregate tumor copy number can be described as:

Calculation for inference of copy number:

$$Tumor\ copy\ number = \frac{2x(copy\ number\ ratio) - 2x(non\ tumor\ derived\ cfDNA\ AF)}{ctDNA\ AF}$$

This formulation uses the measured copy number ratio of the sample, along with the mean tumor fraction of the sample, to calculate the copy number state in tumor cells. This calculation is important, as it accounts for differences in the tumor content between samples, which can vary widely in circulating tumor DNA.

The inferred copy number also highlights the clonal heterogeneity of multiple myeloma. However, interestingly, we found that the dominant clones have monoallelic loss in 4 out of 5 patients. One patient (CAR-04) likely had a dominant clone with biallelic BCMA loss. Of note, this patient had prior allogeneic BCMA-targeted CAR T-cell therapy, and he progressed on day 42 days after ide-cel infusion.

Table R1 (Table S5 in manuscript) BCMA expression in the bone marrow and BCMA copy number estimated by CAPP-Seq

	BCMA expression by bone marrow IHC prior to CAR-T	Copy Number estimated by CAPP-Seq	Inferred Tumor Copy Number of the dominant clone
CAR-04	80% positive, 20% dim - negative	0.05	0
CAR-15	Not evaluable	0.65	1
CAR-10	100% positive	1.2	1
CAR-08	100% positive	0.72	1
CAR-03	100% positive (predominantly perinuclear and dim cytoplasmic)	0.51	1

2. Did these 5 patients have other high-risk features like del 17p?

Among these 5 cases we identified as having BCMA copy number loss prior to treatment, 2 of them also had del(17p). Other FISH data for these 5 cases is shown below. When compared to the remainder of our cohort, the rate of high-risk cytogenetics was not different between these cases with BCMA loss and cases without BCMA loss (47% vs 60%, p=0.29). This suggests that

we are not simply finding 'adverse risk' cases, but the loss of BCMA is driving resistance to therapy.

Table R2 Cytogenetics of 5 cases where pre-treatment BCMA loss was observed

	del(17p)	t(4;14)	t(14;16)	gain(1q)
CAR-04	1	1	0	1
CAR-15	0	0	0	0
CAR-10	1	0	0	1
CAR-08	0	0	NA	1
CAR-03	0	NA	NA	NA

3. I would suggest table 1 be broken down as: 1. Newly Diagnosed; 2) RRMM (non-cellular therapy) and RRMM-BCMA CART if possible. Also please indicate what cytogenetic changes are included as 'high risk' as a footnote.

We appreciate this suggestion and have edited the layout of Table 1 as described above. We have also provided clarification for the cytogenetic changes defined as 'high risk' as a footnote. These details and changes can be found in page 16, Table 1 of the revised manuscript.

4. Does Fig 3E list of all of genes assessed? I did not see TNFRSF17 (sorry if i missed)?

Thank you for the question, and the reviewer is correct; you did not miss this. In the original submission, Fig. 3E displays only coding genes present in a certain threshold of patients as explained in the figure legend. In our resubmission, Fig. 3E shows all genes with coding mutations detected in at least 2 cases; this is outlined clearly in the legend of the manuscript. All genes included in the panel are now included in Fig. S5 of the revised manuscript. We detected 2 cases with TNFRSF17 SNVs mutations, and only one with a coding alteration (the other in an intron). These mutations are shown below in a table (Table R3) for the reviewer.

We note that, unlike the deletions in TNFRSF17/BCMA that the reviewer inquired about above, these SNVs did not appear to be enriched for emergent or functional alterations, where only one coding change was observed.

Fig 3E (reproduced for the reviewer) An oncoprint including coding mutations in our cohort detected in at least 2 patients

Fig S5 (reproduced for the reviewer) An oncoprint including all detected coding mutations in our cohort

Table R3 Detected mutations in TNFRSF17 in the study cohort

	Gene	Variant Class	Variant Type	Chr	Position	Tumor Allele	Ref Allele	Residue Change	AF (%)
CAR-15	TNFRSF17	SNV	intron	16	12061354	A	G	none	49.3
CAR-08	TNFRSF17	SNV	missense	16	12060160	C	T	ILE>THR	3.8

Reviewer #3:

Hosoya et al. present a study analyzing the use of ctDNA in MM patients treated with a range of therapies, including CAR-T.

The main idea of the manuscript is that ctDNA is a less invasive method and comparable to WES/WGS/FISH in BM aspirates, and serves as prognostic/predictive biomarker.

Despite the interesting premise, the article needs revision in order to achieve more rigorous language, and the study itself needs more adequate tests/validations in order to support claims made.

We thank the reviewer for their thorough comments and suggestions to improve the manuscript. Our point-by-point responses are included below.

Here are a few of these points, although this list is not all inclusive:

1-Figure 2A does not match its caption. There is mention of WGS, but caption says DLBCL? Where are the driver/most common MM genes?

We thank the reviewer for their comments here and agree this can be presented more clearly, and agree that the point of this figure was not clear in the initial submission. This panel was included because plasma-based ctDNA and CAPP-Seq are established techniques in non-Hodgkin lymphomas (Scherer et al., *Sci Transl Med.* 2016, PMID 27831904, Kurtz et al., *J Clin Oncol.* 2018, PMID 30125215, Kurtz et al., *Nat Biotechnol.* 2021, PMID 34294911). Targeted NGS approaches for tracking MRD are particularly effective for lymphomas due to the stereotyped occurrence of mutations in focused regions of the genome due to the activity of activation induced deaminase (AICDA). Multiple myeloma has a similar stereotyped distribution of mutations, particularly around immunoglobulin genes and super-enhancer region of BCL6, which therefore makes this an excellent candidate for panel-based targeted NGS for ctDNA-MRD.

However, as this paper is focused on multiple myeloma, we agree that it should not include DLBCL cohort in the main figure. We now modified Fig 2A to only include myeloma WGS data, with frequency of mutations in each binned-region shown in y-axis. We moved DLBCL WGS data to Fig S1.

Fig 2A (cited from the manuscript for the reviewer) A bar plot showing the distribution of single nucleotide variants in multiple myeloma from WGS data. The genome was divided into 1,000-bp bins, and the fraction of patients with a variant in each bin was calculated.

2-Figures 2B,C the comparisons do not make sense. Why compare number of SNVs between different cancers?

We appreciate the opportunity to clarify this point. One of the key challenges in tracking tumor from cell-free DNA is the low amount of total cell-free DNA in the blood plasma. Because of this, nearly all modern approaches to tracking ctDNA-based MRD from the plasma rely on tracking multiple mutations for any individual case, in order to increase sensitivity. The main advantage of CAPP-Seq assay is that we can track many mutations simultaneously for sensitive detection to increase sensitivity, all while using a small sequencing panel. Thus, Fig 2B and C do not show the number of SNVs between different types of cancers, but instead show the number of SNVs that can be tracked in individual cases of differing cancer types using small CAPP-Seq panels for ctDNA detection. This emphasizes that multiple myeloma is a disease where the distribution of SNVs is favorable for ctDNA-based detection, as the number of SNVs that you can track using a small panel in the average case is higher than other cancers.

Importantly, the fact that increasing the number of tracked SNVs greatly increases the sensitivity of the ctDNA-based assays has been well described by multiple groups, which has been described previously (Newman et al, Nat Med 2014, PMID 24705333; Wan et al., Sci Transl Med 2020, PMID 32554709; Zviran et al., Nat Med 2020, PMID 32483360). Cancers listed in Fig 2B and 2C have either FDA-approved clinical ctDNA assays or have been vigorously extensively studied. The data presented here on lung cancer, esophageal cancer and DLBCL comes from prior papers (Azad et al., Gastroenterology 2020, PMID 31711920; Chabon et al., Nature 2020, PMID 32269342; Sworder et al., Cancer Cell 2023, PMID 36584673), and are used to demonstrate that the feasibility of tracking ctDNA-MRD is favorable or comparable compared to these other cancers, where ctDNA-MRD is more established. The data comes from prior studies using CAPP-Seq in these cancers reported from our group. We are comparing the number of detectable SNVs and the ctDNA tumor fraction between MM and these cancers in order to justify the feasibility of developing a ctDNA assay in MM. As shown in Fig 2B, CAPP-Seq assay we developed detected adequate number of mutations for tumor tracking, which supports the feasibility of a MM ctDNA assay.

However, we do acknowledge that the above points that we have made for the reviewer were not clearly stated in the original manuscript as submitted. We therefore significantly expanded upon the description for the purpose of these sections and figures of the manuscript in the revised submission. These extensions can be found on page 3 and the legend of Fig 2B and 2C of the revised manuscript.

3-Figure 2D, how many of these mutations in teal are not part of the STAMP panel? How do authors know these mutations are real (ground truth)? How to know the overlap of mutations detected in both assays?

Thank you for this question and the opportunity to clarify the purpose of the comparison originally presented in Figure 2D. The purposes of comparing our MM-focused CAPP-Seq panel and the clinical genotyping Heme-STAMP panel was not to compare the accuracy of genotyping calls, but instead to show how many more mutations can be captured in a small, targeted sequencing panel that is optimized not to capture 'coding' alterations, but instead to capture as many mutations as possible, including intronic and intergenic regions of the genome. As described in the response to R3's question #2 above, tracking as many mutations as possible increases the sensitivity for ctDNA-MRD, which is essential for these methods.

We are also happy to answer the reviewer's question here, regarding the overlap between our MM-CAPP-Seq panel and HemeSTAMP. The genes covered by our myeloma CAPP-Seq panel and the in-house Stanford Heme-STAMP genotyping panel are different, though there is some considerable overlap. The Heme-STAMP panel is tailored towards pan-hematological malignancies and covers 164 genes. Our targeted CAPP-Seq panel is tailored specifically to detect variants in multiple myeloma, thus focused on genes of interest and regions frequently mutated in MM, covering portions of 82 genes. In total, the overlap of these panels is 38 genes, covering 89.9 kb of genomic space.

We assessed whether genes mutations in genes detected by Heme-STAMP were also detected by our CAPP-Seq panel. Among 11 cases where Heme-STAMP data was available, there were a total of 12 mutations detected, which genes are also covered by our CAPP-Seq panel. Among these mutations, 11 out of 12 (92%) were detected by our CAPP-Seq assay. This concordance suggests a high accuracy of our CAPP-Seq approach from cell-free DNA, compared to a clinically reported, targeted sequencing that we use as a gold standard. These mutations are shown below in Table R4 (Table S2 in supplemental manuscript), provided for the reviewer.

Table R4 (Table S2 in supplement) Detected SNVs by Heme-STAMP

	Gene	Chr	Position	VAF (%)	Detected by CAPP-Seq
MM-07	ATM	11	108201117	37	1
MM-11	DNMT3A	2	25464578	10	1
MM-24	PRDM1	6	106553257	48	1
MM-25	KRAS	12	25380275	1	0

MM-27	CCND1	11	69456230	10	1
MM-27	MYC	8	128752641	6	1
MM-30	EGR1	5	137801708	12	1
MM-30	EGR1	5	137801642	10	1
MM-31	NRAS	1	115256529	10	1
MM-31	PPM1D	17	58740530	49	1
MM-31	CCND1	11	69456211	15	1
MM-31	IRF4	6	394972	7	1

Regarding the reviewer's question involving whether mutations identified in the plasma can be confidently determined as tumor-derived somatic variants versus artifacts, we recognize that in any sequencing study there is always concern for artifacts. However, a number of safe guards against this have been taken. These include the fact that 1) our sequencing data is always performed in conjunction with a matched normal; 2) our data utilizes an additional panel of healthy controls (n=26), which accounts for stereotyped artifactual mutations; 3) our methodology uses robust error suppression including both molecular barcoding and in silico error suppression as described in Newman et al., Nat. Biotechnol. 2016, PMID 27018799; and 4) we apply stringent tumor AF thresholds, and do not call mutations below 0.3% AF, which, given our median sequencing depth of 2628, is well within the norm for cell-free DNA sequencing studies. Furthermore, all data analysis was performed in the same manner for both blood plasma and bone marrow samples.

These details were added to the supplemental methods section in the text (page 2 of supplemental material):

“somatic single nucleotide variants (SNVs) were called using a custom variant algorithm optimized for cfDNA as previously described. Briefly, the sequencing data undergoes vigorous filtering including 1) matched normal to account for germline mutations, 2) a panel of healthy controls to account for stereotyped artifactual mutations, 3) molecular barcoding and in silico error suppression to account for sequencing.”

4-Figure 2E-what is unit of limit of detection? In text authors mention coding/noncoding mutations, does it mean coding and non-protein coding genes, or synonymous vs. non-synonymous mutations?

We apologize that this unit of limit of detection was not included in our original submission. In Fig 2E, the limit of detection refers to the lowest concentration of tumor-derived mutated DNA molecules at which there is a 95% probability of detection. The unit is 'tumor fraction', measured as the mean tumor fraction across all tracked variants, as described in Newman et al, Nature Med 2014 and Newman et al, Nat Biotechnol 2016. We have updated Fig 2E to clarify this (pasted below), which has improved the interpretability of the manuscript.

Fig 2E (reproduced for the reviewer) The ctDNA detection limits are shown for coding genes only and coding + non-coding genomic regions based on the number of mutations detected in our cohort.

We also appreciate the opportunity to clarify the terms used. Coding mutations are used to refer to non-synonymous gene mutations, whereas non-coding mutations are used to refer to both synonymous gene mutations and mutations in non-coding or intergenic regions (i.e., DNA changes that do not lead to a change in the amino acid coding sequence of an expressed protein). We have clarified this definition in the manuscript, as seen on page 19 of the revised manuscript.

5-Figure 3A: How many of which? How many overlapped (same patient same day)? What is genotyping? WGS?

Thank you for the opportunity to clarify this point. In Fig 3A we are showing the number of mutations detected per patient, as genotyped in our MM-CAPP-Seq panel. Genotyping was performed by our CAPP-Seq assay on either a plasma or bone marrow sample. The colors are used to show non-coding vs. coding mutations, with the definition of coding used here described in the response to R3Q4 above. We have updated the figure to clarify that each bar represents a single case, and also added the n for the number of patients considered, genotyped from either cfDNA or Bone marrow, to the x-axis.

Regarding the mutational overlap between plasma and bone marrow at paired time-points, this is presented in a different figure. In the revised manuscript, this can be seen in Figures 3F, S6, and S7 in the revised manuscript. We note an overall high degree of mutational overlap, with some differences between these two compartments. This result is as expected, and is consistent with tumor-versus-plasma concordance exercises from prior papers in other histologies (Scherer et al., *Sci Transl Med.* 2016, PMID 27831904; Newman et al., *Nat Biotechnol.* 2016, PMID 27018799; Rossi et al., *Blood* 2017, PMID 28096087).

Fig 3F (reproduced for the reviewer) Venn diagram showing concordance of single nucleotide variants detected from bone marrow (BM) and plasma across 22 paired samples.

Fig S6 (reproduced for the reviewer) Scatter plots of allele frequencies (AFs) of coding genes (red) and non-coding regions (grey) detected from bone marrow (BM) and plasma in patients who had paired samples ($n=22$) ordered by mean plasma variant allele frequency (VAF). Correlation analyses were performed using Spearman's rank-order test.

Fig S7 (reproduced for the reviewer) (A) A bar plot showing the concordance of SNVs detected in the blood plasma and paired BM. The plot shows BM SNVs per case; the bottom panel shows BM SNVs, where the bar is colored dark purple if they are also detected in the plasma. The mean AF of the plasma is shown as heat above the plot. (B) The rate of shared mutations between bone marrow and plasma are shown. Genes with mutations in at least 4 cases are shown.

6-Statement on line 145, regarding Figure 3D: What is the rationale for this statement? Is this a general observation or anecdotal? Coding vs. non-coding genes or synonymous vs. non-synonymous mutations?

We apologize again for our lack of clarity in defining these mutation types. As mentioned above in #3, coding mutations refer to non-synonymous gene mutations, such as NRAS, XBP1, and EGR1 in the figure, while non-coding mutations refer to synonymous gene mutations or mutations in introns or intergenic regions. Fig 3D presents a single exemplary case, which along

with Fig 3B is used to highlight the rationale for tracking non-coding mutations for increased sensitivity in residual disease detection.

Fig 3D (reproduced for the reviewer) An example of tracking coding and non-coding mutations for disease monitoring in a case. Red arrows pointing to timepoints when coding genes became undetectable but non-coding regions and immunoglobulin loci remained detectable.

Fig 3B (reproduced for the reviewer) A pie-chart showing the breakdown of total 5145 SNVs detected in Fig. 3A. *IGH*, immunoglobulin heavy chain; *IGK*, immunoglobulin kappa light chain; *IGL*, immunoglobulin lambda light chain.

Additionally, another case presented in Fig R5 also highlights challenge in current assay in detecting and following driver mutations in the bone marrow. This case shows that the VAFs decreased after CAR-T therapy on day 28 in the bone marrow, however blood plasma detected more mutations that were expanding, which led to a disease relapse. Monitoring disease with comprehensive panel via samples that potentially represent the whole tumor is crucial.

Figure R5 (Fig S3 in manuscript) SNV dynamics of ctDNA vs bone marrow during CAR-T therapy in CAR-06, who relapsed on day 82. (A) Dynamics of each SNV in the plasma and bone marrow. Orange dots are SNVs detected in the BM at VAF > 0.3%, teal dots are SNVs detected only in the plasma at VAF > 0.3%. (B) A scatter plot of allele frequencies of each SNVs in the plasma vs bone marrow on day 28, demonstrating that many mutations were detected at much higher VAF in the blood plasma compared to bone marrow. (C) Copy number plot of plasma vs bone marrow on day 28 across the whole genome, showing that copy number alterations were more readily detectable from the blood plasma.

In our cohort, we sequenced 281 cfDNA samples. Among these, 218 were positive for disease by our methodology that tracks mutations in both coding genes and non-coding regions. When restricting our analysis to only coding mutations in driver genes as defined in Bolli et al. (Nat. Commun. 2014, PMID 24429703), only 140 (64%) remained positive. This discrepancy demonstrates the importance of tracking numerous mutations in both coding and non-coding regions, not just driver mutations, for increasing the sensitivity of the assay.

This becomes particularly pertinent in the setting of minimal residual disease, as shown in Fig R6. In our cohort we had 16 cfDNA samples with paired clinical bone marrow MRD, of which 9 (50%) were MRD+ by our current methodology and of which only 6 (38%) were MRD+ when only tracking driver variants.

Figure R6 MRD detection in patients who had both clinical bone marrow MRD assay and ctDNA assay (n=16). i) Among 16 patients, 7 patients (44%) had positive clinical bone marrow MRD, ii) Among 16 patients, 8 patients (50%) had positive ctDNA-MRD, iii) If we limit our gene panel to only driver genes, then only 6 patients (38%) were ctDNA-MRD positive.

7-Please provide a version of 3E for conventional WES/WGS from marrow for comparison purposes.

Thank you for the chance to clarify this figure and our manuscript. The purpose of our manuscript, and indeed of applying CAPP-Seq to MM, is not to replace WES or WGS, but to enable mutational genotyping and MRD detection from cell-free DNA. The data presented in Fig 3E presents the mutational data from our MM- CAPP-Seq assay, and therefore is limited to only genes and genomic regions represented in our CAPP-Seq panel. WES and WGS were not performed on these cases for two reasons: 1) most of our cases were genotyped directly from plasma, as described in our manuscript, which is not amenable to deep genotyping using WES or WGS, which is typically performed on purified tumor specimens, and 2) this is beyond the scope of our manuscript. We are therefore unable to provide this data from the exact same cases for comparison purposes.

Importantly, while WES/WGS have been used conventionally to identify genomic alterations of MM in research studies, these techniques are not routinely used in the clinic for genotyping or for mutational tracking.

It has been challenging to apply for sensitive tracking of mutations due to low depth of sequencing, Thus detecting mutations from WES/WGS data not within the scope of this manuscript. However, we are happy to be able to compare the results from our targeted sequencing data to existing, published results using WES and WGS in MM. This allows us to compare similar results to published WES/WGS data in mutational frequency of common genes that are covered by both our panel and WES/WGS, such as KRAS, NRAS, BRAF, and TP53, as is described in the main text. Importantly, our approach yields very similar rates of mutations in these key genes compared to prior studies, as described on page 4 of the revised manuscript.

We also have included a figure below (Fig R7) comparing the frequency of commonly mutated genes in our MM CAPP-Seq cohort to other studies with conventional WES sequencing (Walker et al., JCO 2015, PMID 26282654; Lohr et al., Cancer Cell 2014, PMID 24434212; Bolli et al., Nat. Commun. 2014, PMID 24429703). Of note, the rate of *TP53* mutation was significantly higher than Walker et al.³⁹, however this likely reflects the fact that Walker et al. only included NDMM, and our cohort contains RRMM. This figure is also presented in the supplement of the revised manuscript, as Figure S4.

Figure R7 (Fig S4 in manuscript) Fraction of patients with main driver mutations detected by WES vs CAPP-Seq.

In addition to this comparison, between our data and published WES/WGS data, we also compared plasma-based genotyping and bone-marrow-based genotyping (ie, tumor genotyping) in cases in our cohort where both samples were available. This data is shown in both Figure S6 and S7 of the revised manuscript, and reproduced here for the reviewer's convenience. Importantly, as shown in Figure S7, the AF of alleles seen in tumor (x-axis) and in plasma (y-axis) are significantly correlated in the majority of cases. Furthermore, as seen in Figure S7A, the majority of alleles seen in the tumor are also observed in the plasma for cases with ctDNA > 0.1%, further demonstrating the robustness of our assay.

Fig S6 (reproduced for the reviewer) Scatter plots of allele frequencies (AFs) of coding genes (red) and non-coding regions (grey) detected from bone marrow (BM) and plasma in patients who had paired samples (n=22) ordered by mean plasma variant allele frequency (VAF). Correlation analyses were performed using Spearman's rank-order test.

Fig S7 (reproduced for the reviewer) (A) A bar plot showing the concordance of SNVs detected in the blood plasma and paired BM. The plot shows BM SNVs per case; the bottom panel shows BM SNVs, where the bar is colored dark purple if they are also detected in the plasma. The mean AF of the plasma is shown as heat above the plot. (B) The rate of shared mutations between bone marrow and plasma are shown. Genes with mutations in at least 4 cases are shown.

8-Line 158, re: Fig S1, VAFs seem to be biased towards detection in BM but not in plasma. Meaning that there is a lower detection from ctDNA, in many cases, of all genes. Venn diagram dimensions are misleading (40% of area of BM detected SNVs should be outside the intersection). How are the ones only detected in ctDNA considered real or artifact?

The reviewer is correct that in some cases, detection of tumor alleles is seen at higher levels in the bone marrow than in the plasma. This is true in both the allele fraction of somatic mutation, and the number of mutations identified. This is not surprising, as the purity of tumor DNA in ctDNA is quite low, as across the range of cancer histologies (Bettegowda et al, Sci Transl Med.

2014, PMID 24553385). Sequencing tumor tissue DNA (i.e., BM-derived DNA), on the other hand, results in highly pure tumor DNA. Moreover, for our study, when we had bone marrow available, we utilized CD138-sorting of the bone marrow cells whenever feasible, thereby even further enriching tumor DNA. Thus, we expect to have a higher VAF in the bone marrow compared to than blood plasma, as the tumor cells were enriched from the bone marrow, and low tumor DNA concentrations in ctDNA is a known challenge in the field of liquid biopsies.

However, we respectfully disagree that this is an issue with our study. Indeed, the purpose of ctDNA is not to become 'superior' to invasive tissue genotyping for identifying mutations. It is to be a less invasive / non-invasive method for identifying and tracking mutations and MRD, thereby alleviating the need for biopsies. Indeed, the high correlations seen between the bone marrow and blood plasma compartments in Figure S7 *despite* the lower allele fractions in plasma demonstrates the utility of this approach, as a blood draw is easier to perform, and less painful, than a bone marrow biopsy (please see the response to Reviewer 1 as well).

With regards to the reviewer's comment on the Venn diagram from Fig 3F in the original submission, this was in fact created with proportional areas proportional to the number of SNVs in each category. However, we do recognize that proportional areas for circles are challenging to interpret. Therefore, for clarity we have replaced Fig 3F with a bar chart.

Fig 3F (reproduced for the reviewer) Venn diagram showing concordance of single nucleotide variants detected from bone marrow (BM) and plasma across 22 paired samples.

Regarding the reviewer's question involving whether mutations identified in the plasma can be confidently determined as tumor-derived somatic variants versus artifacts, we recognize that in any sequencing study there is always concern for artifacts. However, as we replied in question 3, a number of safe guards against this have been taken including using matched normal, panel of healthy controls to account for stereotyped artifactual mutations, sequencing error suppression. Additionally, we apply stringent AF thresholds, and do not call mutations below 0.3% AF, which, given our median sequencing depth of 2628, is well within the norm for cell-free DNA sequencing studies. Furthermore, all data analysis was performed in the same manner for both blood plasma and bone marrow samples.

Moreover, our data demonstrates that the mutations that we identify in the blood plasma are highly concordant with those identified in the tumor, when tumor material is available. This is shown both at the level of allele fractions (Figure S6 as above), as well as genotyped variants (Figure S7A as above). Here we will also emphasize that, mutations identified in the plasma were done so de novo i.e., without using prior knowledge from the tumor. Given the high rate of

mutations seen in the tumor that are also recovered in the plasma, with observed correlations in allele fractions, we can have confidence in our genotyping.

Finally, to further demonstrate the fidelity of these 'plasma identified' tumor variants, we provide for the reviewer the following figure R8 below. Perhaps the best way to assess the observed 'plasma identified' mutations is to track these mutations over a course of therapy, and determine if they track along with the mutations seen in the original tumor. The below figure included 3 exemplary cases who underwent CAR-T therapy, showing the dynamics of both shared SNVs between the plasma and tumor (top), and the dynamics of SNVs found solely in the plasma (bottom). The SNVs that are unique to the plasma in the bottom panels track with the overall dynamics of disease response as shown by the top panels.

Figure R8 SNV dynamics of representative 3 cases including shared SNVs between the blood plasma and bone marrow (top) as well as SNVs unique to the blood plasma (bottom).

9-Line 160, re: Fig S2A: These comparisons seem to be in overall number of mutations in the cohort, instead of determining agreement between two assays in each patient. Please address this.

Thank you for bringing this to our attention and giving us the opportunity to clarify. Fig S2A compares SNV detection between bone marrow and blood plasma from each case at a paired time point. The lower plot displays overall SNV concordance, with dark purple showing shared SNVs between the bone marrow and plasma, and light color showing SNVs that were detected only in the bone marrow. The upper plot shows the number of SNVs unique to plasma, demonstrating that increasing variant allele fraction and tumor burden in the plasma leads to higher concordance between the bone marrow and plasma compartments.

However, we agree that the figure has complex information without homing the point, thus we have edited the figure to remove this upper plot. The edited figure (Fig. S7A) is shown below.

Fig S7A (reproduced for the reviewer) A bar plot showing the concordance of SNVs detected in the blood plasma and paired BM. The plot shows BM SNVs per case; the bottom panel shows BM SNVs, where the bar is colored dark purple if they are also detected in the plasma. The mean AF of the plasma is shown as heat above the plot.

10-Line 162: What does this sentence mean?

Line 162: *“This suggests that in most cases of MM, plasma genotyping is comparable to invasive tumor mutational genotyping.”*

This sentence is in reference to Fig S2A (Fig S7A in revised manuscript as above), which shows high concordance in SNV detection between plasma and bone marrow above a certain plasma VAF threshold. We have revised this threshold to 0.3%, above which the sensitivity for detection BM-derived SNVs was 82%. This new threshold is consistent with our prior analyses as described in the *Sequencing Data Analysis and Variant Calling* section in the supplement.

We have revised the prior statement for clarity in page 5 of the main manuscript:

“The frequency of overlapping mutations suggests that, in most cases of MM, where the ctDNA AF is above the threshold of detection, BM-derived SNVs can be reliably detected from the plasma.”

11-Line 165: Or maybe they are artifacts? Or derived from other cancers these patients may have? Is there a ground truth to compare to?

This question is similar to the question raised by the reviewer in question 8 above. As we discussed in reply to question 8, The reviewer raises an important point, which is the issue of artifacts in sequencing studies. As discussed above in #8, our sequencing data undergoes a rigorous filtering process to remove germline-derived SNVs, clonal hematopoiesis seen at low allele fractions, recurrent stereotyped artifacts, SNPs by using healthy controls and sequencing errors using a number of methods described in reply to question 8. As we outlined in response to that question, the fact that de novo variant identification from the plasma yields a high rate of recovery and concordance with gold-standard tumor-derived genotypes, we have high

confidence that our mutations presented in this manuscript are not artifacts. While we cannot exclude second malignancies, we do note that these patients did not have second cancers, after careful review of their clinical histories. Furthermore, as shown in the reply to question 8 above, the mutations identified in the plasma-only tracked the patient's tumor response faithfully, showing the same tumor dynamics as the shared alleles seen in both the bone-marrow and plasma compartments. The fact that, in these presented cases, the variants identified in the plasma-only also track the tumor, suggests that these are indeed tumor alleles (please see the reply to question 8 above). Additionally, our SNV detection methodology is applied in the same manner to both bone marrow and blood plasma compartments.

12-Line 166: Because they are more frequent and more clonal? Most mutations in MM are low frequency, apart from these, and given the limited cohort size of this study, it is expected that only these mutations are found in a reasonable number.

The reviewer is referencing line 166: *"Additionally, mutations in driver genes such as NRAS, KRAS and TP53 were more likely to be shared between BM and plasma when compared with other mutations (Fig. S2B)."*

This comment is regarding Fig S2B (S7B in the revised manuscript as below), which shows the rate of shared mutations between the bone marrow and plasma compartments in patients with both samples at paired timepoints. As the reviewer correctly points out, NRAS, KRAS, and TP53 are the most commonly shared between these two compartments, likely because they are the most truncal. We have clarified this comment in the revised manuscript to reflect the fact that these most recurrently mutated genes are likely the most clonal. It is now modified in page 5 as: *"Additionally, mutations in driver genes such as NRAS, KRAS and TP53 were more likely to be shared between BM and plasma when compared with other mutations, suggesting these mutations may occur in early pathogenesis and more clonal."*

Notably, the genes shown are only those with mutations detected in at least 4/22 (18%) patients, and thus we are excluding genes with very low mutational frequency. We have added the number of patients with mutations to the figure for clarity.

Fig S7B (reproduced for the reviewer) The rate of shared mutations between bone marrow and plasma are shown. Genes with mutations in at least 4 cases are shown.

13-Line 176, re:" Fig S3A: Unfortunately these CNA results are not very convincing nor impressive. It seems it cannot replace regular FISH, given does not do translocations, thus BM aspirate remains necessary.

We appreciate the reviewer's comments on our copy number analysis, and the opportunity to clarify the goal of this study, as well as the current state of the art regarding CNV analysis from ctDNA. The intention of this study is to develop a liquid biopsy tool to non-invasively (ie, through a blood draw) 1) monitor residual disease in patients with MM, as well as 2) assess ongoing mutational processes and resistance mechanisms to therapy as patients undergo treatment. The goal of our study, and indeed our assay, is not to replace bone marrow biopsies in routine clinical practice today. As we envision using this assay in the clinic in the future, it would serve as a supplement – not as a replacement – to bone marrow aspirates, to provide a non-invasive method for disease monitoring. The primary use of our ctDNA assay is for highly sensitive SNV tracking, and we argue that the novelty of our method is in the ability to receive multi-faceted sequencing data from both SNVs and CNAs, as well as CAR-ctDNA. We have added the following text to our manuscript to highlight, and clarify, the purpose of our study and this assay, as well as the importance that this is meant to supplement current molecular profiling with FISH through bone marrow aspirates:

(Page 11 of main manuscript)

“we envision ctDNA assays will supplement – rather than replace – current bone marrow assays, which will remain important to establish a pathological diagnosis and to assess structural alterations such as CNAs and translocations using gold-standard FISH. The strengths of our approach are to non-invasively characterize MM, to monitor residual disease, and to assess resistance mechanisms to therapy as patients undergo treatment.”

Regarding the performance of our assay to capture gold standard copy number alterations, we used clinical bone marrow FISH as gold standard methods to compare our ctDNA-based copy number assay. In our cohort, only 29% of the patients had FISH test within one month of sample collection. Thus for 71% of the patients, we used historical FISH data, which underscores the fact that in many cases, FISH testing is usually not repeated upon relapse. The fact that many patients do not have FISH on a bone marrow aspirate repeated at the time of relapsed disease further highlights the need for a non-invasive, blood-based approach.

This difference in timepoints may reflect the sensitivity. Moreover, we would like to highlight that sensitive detection of copy number alterations from ctDNA remains a major challenge in the field. The performance of our approach is on par with, even including the historical FISH data as 'gold standard', the sensitivity is on par with current ctDNA copy number methods, with a limit of detection between 1-3% (Wang et al., *Brief Bioinform.*, 2022, PMID: 36056740; Mosquera et al., *Blood*. 2022;140(Supplement 1):10712-10713; Adalsteinsson et al. *Nat Commun.* 2017, PMID: 29109393).

Moreover, while the sensitivity of our approach is imperfect, the very results of our paper demonstrate that despite this, CNA analysis from ctDNA can lead to important biological findings. Indeed, the detection of emergent copy number alterations, including deletions in TNFRSF17/BCMA, demonstrates that our method can, in fact, detect clinically significant alterations. While bone marrow aspirates would also be able to detect these, as stated above, many patients do not get repeated bone marrow aspirates to assess for emergent alterations. Furthermore, FISH does not provide a broad view of CNAs, but only an assessment of a limited number of targets.

As the reviewer has drawn attention to, translocations are an important prognostic marker in multiple myeloma, which is out of scope of this manuscript. We added in the text as a limitation of this study in the Discussion in page 11; “ *we did not explore detection of translocations using our approach, which are a key recurrent type of alteration in MM.*”

14-Section starting on line 206: ctDNA collected 28 days after therapy initiation (please provide details on treatment of non-CAR-T patients) is hardly a predictive biomarker, unless the ctDNA were collected before initiation of therapy. Please compare fold change of M-Spike/SFLC to determine that ctDNA is superior.

Thank you for this comment. We agree with the reviewer that given the way the term “predictive” is used in the biomarker field to select a therapy for a given patient, our usage of “predictive” is not precise. We have since revised the terminology to “prognostic” such as “*ctDNA dynamics as a prognostic marker for treatment response in myeloma*”.

We provide the treatment information for patients below in Table R5 and in the revised manuscript as Table S3.

Regarding the comparison between M-Spike and ctDNA, in Fig. 4E, we demonstrate that treatment response is correlated with fold change of ctDNA on day 28, regardless of treatment modality (CAR-T or other). We also explored the correlation of treatment response with M-spike/dFLC, which is shown below. We see an overall concordance of treatment response with both M-spike and dFLC fold changes, although importantly, these assessments were not feasible in patients with oligo/non-secretory disease, which was applicable to 11 out of 31 patients (35%) when evaluated by M-spike; 2 out of 31 (6%) when evaluated by FLCs. Additionally, there were many patients with minimal or no change in M-spike or FLC difference that had varying responses to treatment, highlighting the need for an improved method of prognostication. These figures are now included in the manuscript Fig S10.

Table R5 (Table S3 in revised manuscript) Treatment regimens of the cohort in Figure 4F

Hosoya et al. *Deciphering response dynamics and treatment resistance from circulating tumor DNA after CAR T-cells in multiple myeloma.*

ID	Current line of therapy	Treatment
MM-05	6	Venetoclax/carfilzomib/dexamethasone
MM-06	6	Cyclophosphamide/etoposide/cisplatin/dexamethasone
MM-07	1	Daratumumab/dexamethasone
MM-11	17	Selinexor/dexamethasone
MM-15	1	Daratumumab/bortezomib/dexamethasone
MM-19	2	Daratumumab/carfilzomib
MM-20	1	Carfilzomib/lenalidomide/dexamethasone
MM-24	1	Daratumumab/cyclophosphamide/bortezomib/dexamethasone
MM-28	1	Daratumumab/bortezomib/lenalidomide/dexamethasone
MM-32	3	Cyclophosphamide/doxorubicin/cisplatin/etoposide
CAR-01	10	Ide-cel
CAR-02	6	Ide-cel
CAR-03	6	Ide-cel
CAR-04	7	Ide-cel
CAR-05	16	Ide-cel
CAR-06	7	Ide-cel
CAR-07	5	Ide-cel
CAR-08	5	Cilta-cel
CAR-10	5	Ide-cel
CAR-11	10	Ide-cel
CAR-12	9	Ide-cel
CAR-14	8	Ide-cel
CAR-15	9	Ide-cel
CAR-17	12	Ide-cel
CAR-18	6	Cilta-cel
CAR-19	6	Cilta-cel
CAR-20	16	Cilta-cel
CAR-21	5	Cilta-cel
CAR-22	9	Ide-cel
CAR-24	11	Ide-cel
CAR-33	10	Ide-cel

Figure R9 (Fig S10 in revised manuscript) Waterfall plots demonstrating fold change of M-spike and difference between involved and uninvolved free light chain (dFLC) in association with clinical response by IMWG.

Reviewer #4 :

In this study, the authors have developed a circulating tumor DNA approach for the characterization of tumor genomics, monitoring of treatment responses, and early detection of relapses in multiple myeloma. Through the analysis of 352 specimens from 56 patients, both newly diagnosed and those with relapsed/refractory disease, the study establishes not only a correlation between ctDNA and patient outcomes but also extends its application to the tracking of CAR-specific cell-free DNA in patients undergoing anti-BCMA CAR T-cell therapy. Notably, the inverse correlation between ctDNA levels and the relative time to progression post-BCMA-CAR therapy, along with the concordance of ctDNA-MRD with clinical bone marrow MRD, are key findings. Additionally, the capability of ctDNA-MRD to predict clinical relapse and identify resistant clones marks a notable advancement.

The authors greatly appreciate the reviewer's positive comments on the nature of our study and our findings, as well as their suggestions for improvement. We address each of the points below.

However, some areas of the study warrant further clarification.

1. In the study, the authors note that overall, in samples where the ctDNA Allele Frequency (AF) was greater than 0.1%, the sensitivity for detecting bone marrow-derived single nucleotide variants (SNVs) was 82%. However, the study does not explicitly address the rate of false positives associated with this AF level. Clarification on the false positive rate in this context would be a valuable addition.

We appreciate the opportunity to clarify this analysis. We have since updated this statement to a threshold of 0.3% to be consistent with the methodologies described in the manuscript for variant calling. The sensitivity of detecting bone marrow SNVs from the plasma remains 82% at this threshold.

The reviewer also asks an excellent question regarding false positive rate (FPR), as this is an important to discuss when analyzing sequencing data. We recognize that in any sequencing study there is always concern for artifacts. However, a number of safeguards against this have been taken. These include the fact that 1) our sequencing data is always performed in conjunction with a matched normal; 2) our data utilizes an additional panel of healthy controls (n=26), which are utilized to remove stereotyped artifactual mutations; 3) our methodology uses robust error suppression including both molecular barcoding and in silico error suppression as described in Newman et al., Nat. Biotechnol. 2016, PMID 27018799; and 4) we apply stringent tumor AF thresholds, and do not call mutations below 0.3% AF, which, given our median sequencing depth of 2628, is well within the norm for cell-free DNA sequencing studies.

These details were added to the supplemental methods section in the text (page 2 of supplemental material):

somatic single nucleotide variants (SNVs) were called using a custom variant algorithm optimized for cfDNA as previously described. Briefly, the sequencing data undergoes vigorous filtering including 1) matched normal to account for germline mutations, 2) a panel of healthy controls to account for stereotyped artifactual mutations, 3) molecular barcoding and in silico error suppression to account for sequencing.

Additionally, since our last submission, we sequenced an additional 8 healthy control for more robust filtering, now our healthy control includes 26 subjects.

To further describe our specificity and false positive rate of our assay for the reviewer, we describe our methodology in detail below. Our method performs two distinct analyses: i) de novo genotyping of pre-treatment samples without knowledge of tumor-derived SNVs, followed by ii) assessment of residual ctDNA with the knowledge of tumor-derived SNVs. We assessed false positive rate and specificity in both steps.

De novo genotyping

For i), de novo genotyping, we assessed the specificity and FPR by examining the performance of our assay in samples without significant measurable myeloma, using the same genotyping rules we applied to patient samples in our cohort. To do this, we evaluated the detection of new mutations in samples from myeloma patients at time-points of deep remission; given that these patients are in remission, we would not expect to identify myeloma-related variants.

We assessed false positive detection of mutations in a cohort of 23 samples at the time of deep remission. In contrast to subjects genotyped from plasma at the time of active myeloma, the median detected SNVs was 0 (IQ range 0-0.5); cases of active myeloma (n=54) had a median of 86 (IQ range 37.8-107.8) (Fig R10, Fig S2 in supplement material). Overall, across 23 samples, a total of 13 SNVs were identified, all with low allele fraction (median AF 0.74%). This is in contrast to the allele frequencies of SNVs from cases with active myeloma (median AF 2.1%), with 4118 SNVs identified from 54 samples.

Figure R10 (Fig S2 in revised manuscript) Box plot of number of SNVs detected by de novo genotyping in active myeloma versus in remission.

Furthermore, we can evaluate the specificity of our assay at a base-pair level resolution, given the size of our panel. When evaluated across the range of our panel, this represents a specificity of >99.99% across the 540 kilo-bases assessed for identifying a mutation in a single base. Furthermore, of these 13 SNVs, only 1 SNV resulted in coding alteration to a protein-coding gene. Therefore, 22/23 (96%) control samples from patients in deep molecular remissions had no identifiable coding mutation seen from non-invasive blood plasma genotyping, indicating high specificity and low false positive rate of our assay.

We do note that in this analysis, this estimate is a conservative estimate of our specificity and false positive mutation detection. This is because the identified mutations in patients in deep remission could still be true mutations, either from low burden residual disease or clonal hematopoiesis. However, this conservative analysis demonstrates the high specificity and low false positive rate of our assay.

Measuring and tracking ctDNA levels

Regarding false positive rate ii), tracking ctDNA with identified tumor-derived SNVs, we control for the specificity of the assay through a statistical framework comparing the measured signal in known mutated alleles of interest, compared to the background error rate of the panel, through a Monte Carlo framework. This method is described in Newman et al, *Nature Medicine* 2014 (PMID 24705333) and Newman et al, *Nature Biotechnology* 2016 (PMID 27018799). This Monte Carlo framework results in a p-value for the signal of detection in a given sample, for a given list of mutations.

We control for specificity and FPR as follows: we first trained a p-value threshold of ctDNA positivity by assessing false positive rate using SNVs identified from patients in 26 healthy controls using 55 mutation lists, for a total of 1,430 independent tests. This resulted in p-value of 0.031 to keep 5% false positive rate in 26 healthy controls. We then validated the false positive rate in additional 4 withheld healthy controls, which yielded a false positive rate of 5.6% in this withheld validation set, confirming our intended FPR of 5%.

These analyses of FPRs are now included in the main text and supplemental text:

(Page 4 in the main text)

We further evaluated false-positive rate (FPR) and specificity of our assay (Supplemental Methods). As our methods perform i) *de novo* genotyping of pre-treatment samples without knowledge of tumor-derived SNVs, followed by ii) assessment of residual ctDNA with the knowledge of tumor-derived SNVs, we assessed FPR and specificity in both steps. For i), we genotyped samples in remission (n=23) using the same genotyping rules we applied to patient samples in our cohort. In contrast to samples at the time of active myeloma (n=54) with median 86 detected SNVs (median AF 2.1%), samples in remission detected median of 0 SNVs (median AF 0.74%) (Fig. S2). The specificity was evaluated at a base-pair level resolution. Among 23 samples in remission, there were only 13 total SNVs identified, representing a specificity of >99.99% across 560 kilo-bases. For ii), FPR of tracking ctDNA was evaluated in a training cohort of healthy controls (n=26) against SNVs detected from each genotypable patient (n=55; 26 x 55 tests). A p-value threshold for significant ctDNA detection was then set to keep an FPR of 5%. We then used validation cohort (n=4, 4x55 tests), and observed the FPR was 5.6%, further validating 95% specificity of the assay.

(Page 3 in supplement)

Evaluating false-positive rate of de novo genotyping and tumor-informed ctDNA tracking

False-positive rate (FPR) and specificity were assessed as following. As our methods perform two distinct analysis: i) *de novo* genotyping of pre-treatment samples without knowledge of tumor-derived SNVs, followed by ii) assessment of residual ctDNA with the knowledge of tumor-derived SNVs, we assessed FPR and specificity in both steps.

(i) *De novo* genotyping:

For *de novo* genotyping, we assessed FPR and specificity by examining performance of our assay in samples without significant measurable myeloma, using the same genotyping rules we applied to patient samples in our cohort.

Furthermore, we evaluated the specificity of our assay at a base-pair level resolution given the size of our panel.

ii) Measuring and tracking ctDNA levels

FPR of tracking ctDNA was evaluated in a training cohort of healthy controls. Monte-Carlo testing was performed using the detected SNVs from each genotypable patient in the study cohort against controls and false-positive rate was calculated as:

$$\text{false positive rate} = \frac{\text{Number of significantly positive tests}}{\text{Number of tests}}$$

A FPR of 5% was set by adjusting p-value threshold for detection.

We then validated the FPR using 4 additional withheld healthy control plasmas by performing Monte-Carlo testing.

2. Furthermore, the methodology section of the paper indicates that an AF greater than 0.3% was used for analysis. This higher threshold potentially increases the likelihood of false negatives, which in turn might lead to an underestimation of the ctDNA positive rate. Such a discrepancy raises concerns about the accuracy of ctDNA detection and the criteria used to define ctDNA positivity. It would be beneficial for the authors to address this issue, providing insights into their rationale for choosing these specific thresholds and how they impact the study's findings.

As discussed above, a 0.3% AF threshold was used for *de novo* SNV calling. This threshold was chosen as our previous study demonstrated that $AF < \sim 0.2\%$ would substantially increase the sequencing background error (Newman et al., *Nat Biotechnol* 2016, PMID 27018799). By using this threshold, patient-specific SNVs were defined from genotyping sample. We then track these SNVs in subsequent samples by applying Monte-Carlo testing in order to assess ctDNA positivity. During this time, the median limit of detection was calculated as 2.7×10^{-4} as shown in Fig 2E.

Fig 2E (reproduced for the reviewer) The ctDNA detection limits are shown for coding genes only and coding + non-coding genomic regions based on the number of mutations detected in our cohort.

3. The Kaplan-Meier analysis of progression-free survival, stratified by clinical MRD on day 90, offers valuable insights but also raises questions about the biological underpinnings of this particular time frame's effectiveness. Providing a biological rationale for this observation would enrich the study's depth.

This is an important point raised by the reviewer. Day 90 is a clinically significant time point for MM patients receiving CAR T-cell therapy. Currently, there is no standard clinical guideline for obtaining bone marrow MRD after CAR-T, though commonly done at days 60-100 following CAR-T therapy at several centers as the day 30 marrow is often hypocellular and MRD testing may be falsely negative. In line with common clinical practice, at Stanford Cancer Center, where patients for this study were enrolled and treated, patients routinely undergo bone marrow biopsies 90 days after initiating CAR-T therapy to check for evidence of residual disease. The biological basis for this decision is based on a median time to achieve deep response for anti-BCMA CAR-T therapies. The median time to achieve greater than or equal to CR was reported as 2.8 months for idecel (Munshi et al., NEJM 2021, PMID 33626253) and the median time to best response was reported as 2.6 months for cilta-cel (Berdeja et al., Lancet 2021, PMID 34175021).

It would be very interesting to compare clinical MRD and ctDNA-MRD at a different time point, such as day 28, however clinical MRD data was not collected at that time point. One reason for this is that the bone marrow is often hypocellular at day 28, meaning bone marrow MRD cannot be reliably assessed. However, our ctDNA assay does not suffer from this same issue, meaning MRD status can be attained non-invasively at multiple timepoints throughout the course of therapy.

4. The authors of the study make a comparison between their method and earlier methods referenced in sources 16 to 22. However, this comparison is quite brief. A more detailed analysis highlighting how their method improves upon or differs from these previous methods would help readers better understand the progress made in this study.

We appreciate this suggestion to better contextualize our study, and we have since edited the main text with additional comparison, which we have also included below.

(Page 10 of main text)

Prior studies have demonstrated the feasibility and utility of liquid biopsy in MM via ctDNA detection. However, these early approaches have had challenges in detection of low-level MRD due to limited sensitivity.

Table R6 Brief review of literature of liquid biopsy in multiple myeloma

Methodology	Target	Samples	Advantages	Disadvantages	Ref
LP-WGS, WES	Whole genome, whole exome	cfDNA	Characterization of SNVs and CNVs	Requires tumor fraction of >5% for cfDNA detection	Guo et al., Leukemia , 2022
ULP-WGS, WES	Whole genome, whole exome	CTC, cfDNA	Characterization of SNVs and CNVs. Comparison of CTC and cfDNA	Requires tumor fraction of >=5-10% for WES analysis	Manier et al., Nat Commun , 2018
CAPP-Seq	14 genes	cfDNA	Sensitive detection of targeted genes	Limited gene panel	Gerber et al., Hematologica , 2018
ddPCR	9 genes	cfDNA	Sensitive detection of targeted genes	Limited gene panel	Mithraprabhu et al., Leukemia , 2017
CAPP-Seq	5 genes	cfDNA	Sensitive detection of targeted genes	Limited gene panel	Kis et al., Nat Commun , 2017
ASO-PCR	IGH, IGK	PBMC (DNA, RNA), cfDNA	Sensitive detection of MRD	Limited gene panel	Vij et al., Clin Lymphoma Myeloma Leuk , 2014

5. The use of a specially designed target bed file, specifically for multiple myeloma (MM), is an important aspect of this study. In Table S1 of the study, the authors list the genes they are focusing on, but they don't provide much information about other significant genomic areas. Providing the full details of the target bed file or the bait sequences used in the study would make the findings more reliable and complete.

The methodology of designing our custom targeted panel has been described thoroughly in the text. Unfortunately, we cannot share the exact positions included in our bed file, but we submitted a list of detected mutations in the repository at the initial submission.

6. Considering the methodological and conclusion similarities with a study published in *Cancer Cell* in 2023 (reference 34), a more detailed discussion on the novelty of the current study and its distinct contributions compared to the prior research is recommended.

Our study builds upon prior work published in *Cancer Cell*, which explored ctDNA and immune microenvironment factors in patients receiving CAR-T for DLBCL. Our study presents the first comprehensive ctDNA assay for MM, whereas ctDNA has been better explored in lymphomas. Additionally, MM is a disease where residual disease is often monitored using bone marrow biopsies, an invasive procedure for patients that can limit the frequency of sampling and may not capture spatial heterogeneity known to exist in this disease. ctDNA presents a novel solution to this problem.

Building upon Sworder et al., we find additional evidence to support ctDNA as a prognostic factor at various time points following CAR-T infusion, as well as a tool for discovering emergent alterations leading to relapse. Notably, we find that peak levels of CAR-cfDNA significantly predict time to progression (Fig 6F), which was not observed in the prior study in DLBCL. This reflects a need for further research into the importance of CAR-T levels following infusion in different diseases.

Fig 6F (reproduced for the reviewer) Forest plot of variable effects on time to progression. Plot depicts hazard ratios calculated by univariate Cox proportional hazards regressions model, with significant values shown in red ($p < 0.05$). Error bars reflect 95% confidence interval.

7. Correcting the truncated title in reference 37 is essential to ensure the accuracy and completeness of the study's citations.

We appreciate the reviewer noticing this mistake in our references, which has since been corrected.

Reviewer #3 (Remarks to the Author):

Missing points originally raised but not addressed by authors:

We apologize that the original revision did not address reviewer's remarks adequately, and appreciate for their thorough comments and suggestions. Our point-by-point responses are included below.

"1-Figure 2A Where are the driver/most common MM genes?"

We thank the reviewer for the opportunity to clarify Figure 2A and how we built our assay. Specifically, our panel aims to do two functions simultaneously:

1. Capture key coding genes in MM that are frequently mutated in MM (Walker et al., Blood 2018; Lohr et al., Cancer Cell 2014; Bolli et al., Nat Commun 2014).
2. Increase the sensitivity for disease detection from ctDNA, which relies on capturing many mutations per case. Here, we did not limit ourselves to exonic and coding regions of the genome, but instead, we used whole genome sequencing (WGS) data (Lohr et al., Cancer Cell 2014). Figure 2A shows the frequency of mutations from the whole genome.

Indeed, the reviewer is correct that the plot in Figure 2A does not highlight the typical driver genes in MM where coding alterations are found (e.g., *KRAS*, *NRAS*, *BRAF*, *TP53*, etc). This is because these previous studies of MM focus on describing *driver mutations* that are found in *exonic, coding* regions of the genome, which, even the most frequently mutated genes are only found in about quarter of the patients (Bolli et al., Nat Commun 2014; Walker et al., Blood 2018; Maura et al., JCO 2024). In contrast, one of the major goals of our assay is to improve the sensitivity for disease detection of MM from ctDNA, and to do this, we leverage not only coding mutations from the exonic regions of the genome, but also mutations from intronic and intergenic regions. This allows us to increase the number of mutations seen per case, which, in turn, increases the sensitivity for ctDNA disease detection, as described in our manuscript and multiple prior studies (Newman et al., Nat Med 2014; Newman et al., Nat Biotechnol 2016; Wan et al., Sci Transl Med 2020; Zviran et al., Nat Med 2020). Specifically, Figure 2A demonstrates that the prevalence of mutations in MM – when assessed from the whole genome, rather than focused exclusively on coding mutations – reveals that the most common regions harboring mutations in MM occur in regions that are not typically thought of as “coding”, such as the immunoglobulin loci (which are not in most exome capture kits), the *BCL6* super-enhancer, etc. These are regions that are well-described to be targets of activation-induced deaminase (AID), the driver of B-cell receptor

diversification that is key in B-cell lymphocyte biology and B-cell development, thus making biological sense to be present in MM cells (despite not being in exonic / coding regions). Indeed, when viewed through the lens of capturing as many mutations as possible for disease monitoring, these regions yield more mutations than traditional “canonical driver mutations”. This result is key to highlight, and we thank the reviewer for pointing this out.

To highlight this, we have now updated Figure 2A to include the locations of the key driver genes in MM, as well as highlighting in the text and the legend that this figure highlights mutations that occur in any region of the genome, rather than focusing on coding regions.

"3-Figure 2D, how many of these mutations in teal are not part of the STAMP panel? How do authors know these mutations are real (ground truth)? How to know the overlap of mutations detected in both assays?" This remains an issue as the authors cannot claim their assay is better because it calls more mutations, since these could be false positives. A correct approach would be to sequence both tumor and ctDNA with both assays.

Thank you for this question, and we agree that ensuring that the mutations and detections coming from our assay – and indeed any assay – are not false positives is essential. In fact, Reviewer 4 in the first round of reviews also focused on the need to confirm that these mutations are not false positive variants, and what has been done to control the False Positive Rate (FPR) (Reviewer 4, Question 1 in the original round of reviews). Here, we directly address this reviewer’s concerns regarding the possibility of false positives, but also refer this reviewer to the answers to Reviewer 4 in the original review for more details.

Regarding the reviewer’s suggestion that “a correct approach would be to sequence both tumor and ctDNA with both assays” to confirm if the mutations seen in our CAPP-Seq assay are “true positives” or “false positives”, we respectfully disagree. This is because the MM-CAPP-Seq and Heme-STAMP cover different regions in the genome – therefore, mutation that are not identified by Heme-STAMP can be due to either i) not being detected by an assay, or ii) not being covered by an assay.

However, we are happy to address these specific reviewer questions here, as well as provide a detailed explanation of how we controlled for false positive mutations below.

Fig 2D (reproduced for the reviewer) Number of SNVs identified by our CAPP-Seq assay comparing to clinical NGS assay (n=11).

Overlap of Heme-STAMP and MM-CAPP-Seq - The purpose of Fig 2D is to compare the number of mutations called by our MM-CAPP-Seq and clinically available next-generation sequencing (NGS)-based assay (Heme-STAMP). Heme-STAMP is a clinically validated assay that is based on NGS using targeted gene panel. It covers ~670 kb genomic regions including 164 genes. This panel has ~83kb overlapping region with MM-CAPP-Seq panel – however, Heme-STAMP is only validated to report on coding alterations. In total, of all the mutations described in this paper and in Figure 2D, Heme-STAMP covers only a portion of the genome, 20.7% (169/817 mutations) identified samples shown in Fig 2D.

When considering the concordance of these two assays in the overlapping regions, we assessed two features. 1) of all the mutations identified by HemeSTAMP, how many were also identified by CAPP-Seq. As described in our original PBPR, 11 /12 (93%) of all mutations originally identified by HemeSTAMP were also identified by CAPP-Seq. 2) Here, we also assessed how many of the mutations identified by CAPP-Seq, were also

seen in the HemeSTAMP assay. We note that to perform this, we searched beyond the clinically reported variants, as HemeSTAMP only reports out the coding variants of their assay. Here, 89.7% of mutations seen with high allele fraction (>3%) by CAPP-Seq were also observed by HemeSTAMP. We do note that while this concordance is very high, it is not 100%, which we strongly suspect is due to the fact that these assays did *not* sequence the exact same sample (i.e., Heme-STAMP sequences bone marrow DNA without matched normal, while the paired CAPP-Seq samples in this analysis are mostly sequenced from cell-free DNA with matched normal). Nevertheless, this high concordance is in line with tumor/plasma concordance analyses in other tumor types, and confirms the accuracy of our assay.

We also would like to directly address the reviewer's suggestion to sequence the same analyte on both assays. Of all of the samples in Fig 2D, only one (MM-17) utilized the exact same sample used for both assays (ie, unsorted bone marrow aspirate). For this sample, we saw near perfect correlation of the allele frequency of each mutation between these assays (Fig R2). This suggests that, when evaluated on the same sample, using the same genomic regions, these tests produce comparable results.

Assessing the false positive-rate / specificity of our assay using “true negative” samples

As sequencing the same samples on both assays does not provide an opportunity to evaluate specificity and “false positives” in the non-overlapping regions, we suggest an alternative, as previously outlined in our initial PBPR to Reviewer 4. We suggested that the use of “known negative samples” – ie, samples that are known to not harbor mutations – can be used to assess the rate of false positives and specificity. The rate of “false positives” can simply be assessed as the rate of identifying mutations in these “known negative samples”, as these must all be false positives by definition. Notably, this is a common approach for assessing the false positive rate and “limit of blank” in Clinical & Laboratory Standards Institute (CLSI) guidelines. As written in the original

reply, we assessed false positive rates for i) de novo genotyping (as per the Reviewers questions here) and ii) SNVs monitoring steps as below.

Assessment of false positive rate of CAPP-Seq assay

To further describe our specificity and false positive rate of our assay for the reviewer, we describe our methodology in detail below. Our method performs two distinct analyses: i) de novo genotyping of pre-treatment samples without knowledge of tumor-derived SNVs, followed by ii) assessment of residual ctDNA with the knowledge of tumor-derived SNVs. We assessed false positive rate and specificity in both steps.

De novo genotyping

For i), de novo genotyping, we assessed the specificity and FPR by examining the performance of our assay in samples without measurable myeloma by clinical or molecular features, using the same genotyping rules we applied to patient samples in our cohort. These samples can be thought of as “true negative samples” – ie, as myeloma is not present to a discernable degree, these samples are expected to yield 0 or near-0 mutations. Detection of mutations in these samples are highly likely to be false positives.

To perform this analysis, we evaluated the detection of new mutations in samples from myeloma patients at time-points of deep remission; given that these patients are in remission, we would not expect to identify myeloma-related variants.

Therefore, we assessed false positive detection of mutations in a cohort of 23 samples at the time of deep remission. In contrast to subjects genotyped from plasma at the time of active myeloma, the median detected SNVs was 0 (IQ range 0-0.5); cases of active myeloma (n=54) had a median of 86 (IQ range 37.8-107.8) (Fig R3, Fig S2 in supplement material). Overall, across 23 samples, a total of 13 SNVs were identified, all with low allele fraction (median AF 0.74%). This is in contrast to the allele frequencies of SNVs from cases with active myeloma (median AF 2.1%), with 4118 SNVs identified from 54 samples.

Furthermore, we directly address the Reviewer’s question regarding the possibility of the 4,118 SNVs identified as patient-specific multiple myeloma variants being false positives. We performed this by assessing the frequency that these 4,118 SNVs were identified (ie, *de novo* genotyped) in these “true negative” samples. Of these, 4,105 SNVs (99.7% of all SNVs) were identified in 0/23 (0%) of true negative samples,

indicating 100% Specificity in the cases evaluated. The remaining 13 SNV (0.3%) were identified in only 1 of 23 true negative samples, indicating a specificity of 22/23 or 95.6% for these remaining SNVs. Thus, conservatively, the specificity of our assay for de novo genotyping, is 4105 / 4118, or 99.7% of SNVs, having 100% specificity in “true negative samples”.

Fig R3 (Fig S2 in revised manuscript) Box plot of number of SNVs detected by de novo genotyping in active myeloma versus in remission.

We further evaluated the specificity of our assay at a base-pair level resolution, given the size of our panel. When evaluated across the range of our panel, this represents a specificity of >99.99% across the 540 kilo-bases assessed for identifying a mutation in a single base. Furthermore, of these 13 SNVs, only 1 SNV resulted in coding alteration to a protein-coding gene. Therefore, 22/23 (96%) control samples from patients in deep molecular remissions had no identifiable coding mutation seen from non-invasive blood plasma genotyping, indicating high specificity and low false positive rate of our assay.

We do note that in this analysis, this estimate is a conservative estimate of our specificity and false positive mutation detection. This is because the identified mutations in patients in deep remission could still be true mutations, either from low burden residual disease or clonal hematopoiesis. However, this conservative analysis demonstrates the high specificity and low false positive rate of our assay.

Measuring and tracking ctDNA levels

In addition to the above assessment of the false positive rate for genotyping SNVs, we also assessed ii), the false positive rate for tracking ctDNA from previously identified tumor-derived SNVs. While this reviewer did not inquire about the false positive rate for this aspect of our study, we do note that Reviewer 4 asked for this material (R4Q1 in the

initial point by point response). If of interest, we encourage the reviewer to assess our response there regarding this point.

We also note that in reply to question 1 from Reviewer 4, we modified the text to include these analyses and description of our assessment of the false positive rate. These can be found in page 3 of Supplemental Methods of the currently submitted manuscript.

"7-Please provide a version of 3E for conventional WES/WGS from marrow for comparison purposes." A true validation of false positives/false negatives is to base on a ground truth. I agree that the goal is not to replace BM-WES, but in order to determine how many new mutations/clones exist/disappear in sequential assays, it is important to determine how accurate assay is. However, the authors are the ones who claimed that their assay in most MM cases was comparable to invasive tumor genotyping, and now have changed their text to "... suggests that in most cases of MM, where the ctDNA 187 amount is above this level, BM-derived SNVs can reliably detected from the plasma".

Thank you for these additional questions and comments. Unfortunately, we cannot provide paired tumor/normal WGS or WES for the entire cohort of this study, as this is beyond the scope of this current study, and tumor samples are not available for most of these cases. In addition to this being beyond the scope of this study, we also note that this is not a feasible experiment to perform, as our study primarily aims to genotype MM samples directly from plasma-derived ctDNA, which has a much lower tumor fraction than tumor samples (particularly lower than CD138+ enriched bone marrow aspirate samples). As described in the previous question, the median allele fraction for tumor alleles from blood plasma in active MM is only 2.1%. Thus, to identify mutations from WGS from these plasma samples, a tumor/normal WGS would need to be performed at $> \sim 250x$ depth in order to achieve at least 5 supporting reads for the typical somatic allele (we use this number as 5 supporting reads would be a quite low number to be able to genotype with any confidence). This is not feasible with modern sequencing at a reasonable cost or throughput, and is indeed the very motivation for targeted deep sequencing in cell-free DNA.

We also point out that WES/WGS is not the only "ground truth" or gold standard for detecting tumor-derived mutations. There are many clinically validated non-WES/WGS tests to assess gene mutations. These include NPM1/IDH1/IDH2 mutation assay for AML, BRCA1/2 mutation assay for breast cancer (RT-PCR based NGS), comprehensive genomic profiling for various cancer (FoundationOne, hybrid capture-based NGS). Additionally, there are many clinically validated, FDA-approved diagnostic tests for mutational profiling using cell-free DNA- these include EGFR assay for lung cancer (FoundationOne Liquid CDx, Guadant360) and BRCA1/2 assay for breast cancer

(FoundationOne Liquid CDx) (website for “List of cleared or approved companion diagnostic devices”: <https://www.fda.gov/medical-devices/in-vitro-diagnostics/list-cleared-or-approved-companion-diagnostic-devices-in-vitro-and-imaging-tools>).

While not the comparison to WGS that the reviewer requests, we do note that, in our submission, we provide two sources of orthogonal “ground truth” – first, a tumor versus cfDNA concordance analysis when using the same sequencing space (ie, MM-CAPP-Seq), as shown in Figure S7, and a comparison to tumor sequencing from the clinically available HemeSTAMP panel, where available, as shown in Table S2.

With all of this being said, we do understand the Reviewer’s preference for WGS as a source to validate our assay against. As noted above, this can only be assessed in the setting of high-purity tumor samples. In our dataset, one such case does exist (CAR-10), where we had a CD138-sorted bone marrow sample with high tumor purity which we performed both WGS (median depth = 5x) and MM-CAPP-Seq. In Figure R4, presented for the reviewer, the allele frequency (AF) of mutations detected by CAPP-Seq (x-axis) are plotted against the AF of the same mutations detected by WGS (y-axis). Notably, the concordance between these two assays is high. For SNVs with WGS depth ≥ 5 , 85.2% (46 out of 54) of mutations with AF $\geq 20\%$ observed by CAPP-Seq were also observed by WGS. We do note that the depth of our WGS is low and thus can miss mutations with low tumor fraction, and lead to some noise around the overall correlation. However, this correlation, as well as high concordance, supports the accuracy of the CAPP-Seq approach (**Fig. R4**).

Fig. R4 Allele frequency of mutations detected by CAPP-Seq, which are plotted against allele frequency assessed by WGS (including only positions where WGS depth ≥ 5).

"11-Line 165: Or maybe they are artifacts? Or derived from other cancers these patients may have? Is there a ground truth to compare to?" Once again, the question is not regarding bioinformatics analysis, but false positives that can emerge from the assay itself. We need a control/groundtruth using another assay (WES/WGS, etc.) for a true validation.

This comment is similar to Q3 from Reviewer 3 above, and we agree with the need to control for false positives, although disagree that WES or WGS are the best ways to do this. As discussed in the replies above, we have extensively controlled for false positives, through the use and evaluation of "true negative samples" which prove the specificity of our approach. We have also demonstrated comparisons to external assay, including the clinically available HemeSTAMP assay, as well as WGS, where available and where tumor purity was sufficiently high, as described above. Furthermore, our additional deep sequencing of matched normal leukocyte DNA for possible germline alleles and clonal hematopoiesis. We strongly believe this is sufficient analysis to confidently state that these are not technical artifacts from the assay itself.

However, it is true that we cannot with 100% absolute certainty state that these mutations are not coming from another tumor (although these patients did not have other malignancies identified). We have modified the text to account for this possibility, as described below.

(Prior text) In addition to BM-derived SNVs, we detected an additional 483 SNVs in the plasma across these 22 patients that were not present in the BM, including *KRAS*, *NRAS*, *TP53*, and *BRAF* mutations, suggesting ctDNA may capture spatial tumor heterogeneity not present in BM biopsies.

(Revised text) In addition to BM-derived SNVs, we detected an additional 483 SNVs in the plasma across these 22 patients that were not present in the BM, including *KRAS*, *NRAS*, *TP53*, and *BRAF* mutations, suggesting ctDNA may capture spatial tumor heterogeneity not present in BM biopsies. While our assay does include deep sequencing of matched normal to exclude germline alleles and clonal hematopoiesis, it is possible these alleles could arise from another malignancy. However, none of these patients had second malignancies documented at the time of assessment, making this an unlikely source of these mutations.

"13-Line 176, re:" Fig S3A: Unfortunately these CNA results are not very convincing nor impressive. It seems it cannot replace regular FISH, given does not do translocations, thus BM aspirate remains necessary." I feel the authors are trying to associate as many features the to product as then can, when instead they should focus on providing strong evidence of its main use and reliability. This is another case where even regular WES/WGS has difficulty in assessing CNVs, and ctDNA with an even more limited gene set, makes it even more challenging. I stand by my original comments.

We agree with the reviewer that assessing CNVs is challenging using WGS/WES. Generally, the consensus is that we would need tumor fraction of at least 3% in cell-free DNA for the assessment of CNVs using WES/ultra low-pass (ULP)-WGS (Adalsteinsson et al., *Nature Communications* 2017). Our CNV analysis (CANARy), as described in the Supplemental Methods, uses both on-target and off-target sequencing reads. Off-target sequencing reads come from the whole genome area with sequencing depth of 0.1-0.3x, which is similar to ULP-WGS. We indeed show that the detection of CNVs by our assay is comparable to prior reports of CNV analysis by ULP-WGS. We also note that the goal of our test is not to replace standard FISH analyses for CNVs, but to allow translational research and discovery, as exemplified by the assessment of *TNFRSF17* deletions (e.g., Fig 6B in the submitted main text). We stand by the utility of our approach for doing so, despite its lesser performance as compared to FISH on a bone marrow aspirate.

However, we agree that it is a fair assessment that NGS – including both targeted and WES/WGS – should not be considered a replacement for the current standard, clinical FISH analysis. We have modified to the text as follows so as not to be misinterpreted by the readers.

(Prior text) For samples with an AF \geq 2.5%, CANARy demonstrated sensitivity of 80, 80, and 100% for detecting del(17p), gain(1q), and del(13q), respectively (Fig. S8B, C). This performance is comparable to dedicated LP-WGS methods from cfDNA, suggesting that CNAs detected using CANARy are concordant with BM FISH tests and that sensitivity improves with higher plasma tumor burden. Additionally, unlike FISH assay, which requires specific probes for chromosome regions of interest, CANARy assesses cfDNA CNAs across the whole genome, leading to discovery of additional alterations.

(Revised text) For samples with an AF \geq 2.5%, CANARy demonstrated sensitivity of 80, 80, and 100% for detecting del(17p), gain(1q), and del(13q), respectively (Fig. S8B, C). This performance is comparable to dedicated LP-WGS methods from cfDNA, suggesting that CNAs detected using CANARy are concordant with BM FISH tests and that sensitivity improves with higher plasma tumor burden. Additionally, unlike FISH assay, which

requires specific probes for chromosome regions of interest, CANARy assesses cfDNA CNAs across the whole genome, leading to discovery of additional alterations. This analysis can be potentially useful in the research and discovery settings; however, we note that this assay should not replace the current gold standard bone marrow-based FISH test for clinical assessment.

"14-Section starting on line 206: ctDNA collected 28 days after therapy initiation (please provide details on treatment of non-CAR-T patients) is hardly a predictive biomarker, unless the ctDNA were collected before initiation of therapy. Please compare fold change of M-Spike/SFLC to determine that ctDNA is superior." Same as above. This claim should not be part of the article.

We addressed reviewer's comment in our prior reply. With the additional reviewer's comments, we modified the title of this subsection from "ctDNA dynamics as a prognostic marker for treatment response in myeloma" to "ctDNA to track treatment response in myeloma".